# Learning perturbation-inducible cell states from observability analysis of transcriptome dynamics

Aqib Hasnain [1] ✉, Shara Balakrishnan[2], Dennis M. Joshy[1], Jen Smith[3], Steven B. Haase [4] & Enoch Yeung[1]

A major challenge in biotechnology and biomanufacturing is the identification of a set of biomarkers for perturbations and metabolites of interest. Here, we develop a data-driven, transcriptome-wide approach to rank perturbation-inducible genes from time-series RNA sequencing data for the discovery of analyte-responsive promoters. This provides a set of biomarkers that act as a proxy for the transcriptional state referred to as cell state. We construct low-dimensional models of gene expression dynamics and rank genes by their ability to capture the perturbation-specific cell state using a novel observability analysis. Using this ranking, we extract 15 analyte-responsive promoters for the organophosphate malathion in the underutilized host organism *Pseudomonas fluorescens* SBW25. We develop synthetic genetic reporters from each analyte-responsive promoter and characterize their response to malathion. Furthermore, we enhance malathion reporting through the aggregation of the response of individual reporters with a synthetic consortium approach, and we exemplify the library's ability to be useful outside the lab by detecting malathion in the environment. The engineered host cell, a living malathion sensor, can be optimized for use in environmental diagnostics while the developed machine learning tool can be applied to discover perturbation-inducible gene expression systems in the compendium of host organisms.

A major step in biomanufacturing and biotherapeutic processes is the optimization of production efficiency and therapeutic efficacy, respectively. Often, destructive or costly measurements such as high-performance liquid chromatography or next-generation sequencing are used to observe the partial or total effect of a compound on known biomarkers that act as proxies for the cellular state. These biomarkers, though difficult to identify, once known, can be used as sensors to gauge the efficiency and efficacy of biotechnological processes across a wide array of experimental conditions.

Transcriptional genetic sensors are a class of biological components that control the activity of promoters[1] and have been used to construct whole-cell (living) biosensors[2–4]. A large portion of transcriptional sensors rely on transcription factor-promoter pairs[5] and have been used in whole-cell biosensing for detection of heavy metals[6], pesticides and herbicides[7–9], waterborne pathogens[10], disease biomarkers[11–14], and many more applications discussed in ref. [15]. Since microbes are found in virtually all terrestrial environments, one could imagine that there would be no shortage of transcriptional genetic sensors for novel sensing applications. However, given a novel sensing application for a target compound or perturbation, transcriptional genetic sensors are typically unknown a priori. Moreover, a complete methodology for discovering sensors and

[1]Department of Mechanical Engineering, University of California Santa Barbara, Santa Barbara, CA, USA. [2]Department of Electrical and Computer Engineering, University of California Santa Barbara, Santa Barbara, CA, USA. [3]California Nanosystems Institute, University of California Santa Barbara, Santa Barbara, CA, USA. [4]Department of Biology, Duke University, Durham, NC, USA. ✉e-mail: aqib@ucsb.edu

biomarkers for the target analyte in novel organisms does not yet exist.

The transcriptional activity of an organism can be measured through RNA sequencing (RNA-seq) to produce a snapshot of the bulk cell state subject to intrinsic and extrinsic perturbations. The typical approach for identifying upregulated and downregulated genes across experimental conditions is to apply differential expression analysis[16,17]. A major pitfall with differential expression analysis is its lack of statistical power when faced with a sparse number of biological replicates. That is to say that the false-positive rate increases drastically when only a small number of biological replicates are available[18] as is often the case due to the costliness of RNA-seq. A related issue arises in that one must sacrifice time points for biological replicates, reducing the fidelity of the dynamical process being studied. As most biological processes are dynamic, time-series profiles are essential for accurate modeling of these processes. Furthermore, differential expression analysis provides no information beyond which genes are upregulated/downregulated[19]. An analysis of expression dynamics provides a potential route to design a sensing scheme for a target analyte for which no single sensor exists.

A typical RNA-seq dataset contains hundreds to tens of thousands of genes; despite that, a subset of genes, often referred to as biomarkers, are typically sufficient for representing the underlying biological variation in the dataset. This is explained by the fact that variations in many genes are not due to the biological process of interest[20] and that many genes have correlated expression levels[21]. Several algorithms to identify the mode-of-action for a compound have been developed from the perspective of network reconstruction and have been used to reconstruct known regulatory networks and discover new ones[22–26]. Network reconstruction relies on steady-state data, is computationally expensive for high-dimensional systems, and the number of unknown parameters necessitate the collection of large, diverse datasets. It is recommended to collect 1/10th the amount of samples as number of genes screened. To screen a model bacteria, e.g. *E. coli*, this amounts to roughly 400 RNA-seq samples; this can be prohibitively expensive. Conversely, we aim to devise a methodology that identifies biomarkers of interest from time-series data that is computationally inexpensive, and we validate our approach on limited datasets by closing the design-build-test loop.

The task of identifying a subset of the state (biomarkers) which recapitulate the entire state (transcriptome/cell state) and explain the variations of interest is well studied in the field of dynamics and controls in the form of optimal filtering and sensor placement[27,28]. In the context of dynamic transcriptional networks, sensor placement is concerned with inferring the underlying cell state based on minimal measurements; this introduces the concept of observability of a dynamical system[29]. The transcriptome is observable if it can be reconstructed from the subset of genes that have been measured. In other words, these genes encode the required information to predict the dynamics of the entire transcriptome. To the best of our knowledge, measures of observability have not been applied to genetic networks to identify genetic sensors, biomarkers, or other key genes.

Due to the lack of DNA-binding information, transcriptional measurements of a population are not sufficient for the identification of biosensors. Several techniques have been developed to analyze temporal correlations in time-series RNA-seq data in order to identify biomarkers of interest[30]. Dynamic clustering tools have been developed which group genes according to co-expression patterns[31,32]. Dynamic gene regulatory network reconstruction (GRN) tools use time-series RNA-seq data to infer the functional interplay among genes when affected by a perturbation[33,34]. In both clustering and GRN, the question of selection of informative genes for downstream targeted gene profiling is not addressed. A primary advantage of observability analysis is the use of temporal correlations to identify an optimal set of biomarkers that act as proxy for the perturbation-induced cell state.

Genes which contribute more to observability are considered as informative genes or optimal biomarkers; these biomarkers can then be selected for targeted gene profiling.

Overall, a systematic approach for identifying genetic reporters from RNA-seq datasets is still an open and challenging issue. In this work, we develop a machine learning methodology to extract numerous endogenous biomarkers for analytes of interest from time-series gene expression data (Fig. 1). Our approach consists of three key steps, each of which is depicted in the middle panel of Fig. 1. The first step adapts dynamic mode decomposition (DMD)[35–37] to learn the transcriptome dynamics from time-series RNA-seq data. Beyond the scope of sensor discovery, we show how the dynamic modes can be utilized to cluster genes by their temporal response. Secondly, we construct and solve a sensor placement problem which assigns weights to each gene[38,39]; highly ranked genes are those which can recapitulate the perturbation-induced cell state. Using this ranking, optimal biomarker genes may be selected. To ensure the ranking is identifying genes which can recapitulate the cell state, the final step is to measure how well a chosen subset of genes can reconstruct the cell state. To validate our proposed methodology, we use our method to generate a library of 15 synthetic genetic reporters for the pesticide malathion[40–42], an organophosphate commonly used for insect control, in the bacterium *Pseudomonas fluorescens* SBW25. The transcriptional sensors play distinct biological roles in their host and exhibit unique malathion response curves. Our method uses no prior knowledge of genes involved in malathion sensing or metabolism. Moreover, we use no data source beyond RNA-seq, thereby providing a cost and computationally efficient approach for biomarker identification.

## Results
### Induction of malathion elicits fast host response
To start, we will first introduce the time-series RNA-seq dataset that we will use throughout this work. The transcriptional activation and repression of the soil microbe *Pseudomonas fluorescens* SBW25 was induced by malathion at a molar concentration of 1.29 µM (425 ng/µL). This concentration was chosen for the following two reasons: (i) it is a moderate amount that can typically be found in streams and ground water after recent pesticide use based on studies done in the United States, Malaysia, China, Japan, and India[43,44], and (ii) the characteristic concentration of a metabolite in bacteria is on the order of 0.1–10 µM[45]. Malathion is an organophosphorus synthetic insecticide used mainly in agricultural settings[46] while SBW25 is a strain of bacteria that colonizes soil, water, and plant surface environments[47]. This makes the soil-dwelling strain a prime candidate for identification of transcriptional genetic reporters for the detection of malathion.

To enable rapid harvesting and instantaneous freezing of cell cultures, we made use of a custom-built vacuum manifold, enabling fast arrest of transcriptional dynamics (Supplementary Fig. 13 and "Methods" section). Following malathion induction, cells were harvested at 10 min intervals for 80 min, obtaining a total of 9 time points across two biological replicates that were sequenced. As the focus of our study is on identifying trends and correlations across time, we heavily favored time points in the trade-off between time points and biological replicates. To identify candidate biomarker genes for malathion induction and subsequently build synthetic transcriptional reporters, we also collected samples from a cell culture that was not induced with malathion. See the "Methods" section for further details on cell culturing and harvesting.

RNA sequencing (RNA-seq) provides a snapshot of the entire transcriptome i.e. the presence and quantity of RNA in a sample at a given moment in time. In this work, we examine the fold change response given by first normalizing the raw counts to obtain transcripts per million (TPM)[48] followed by calculating the fold change of the malathion condition with respect to the negative control, $\mathbf{z} = (\mathbf{x}_M + 1)/(\mathbf{x}_C + 1)$. The implication is that the fold change is the cell

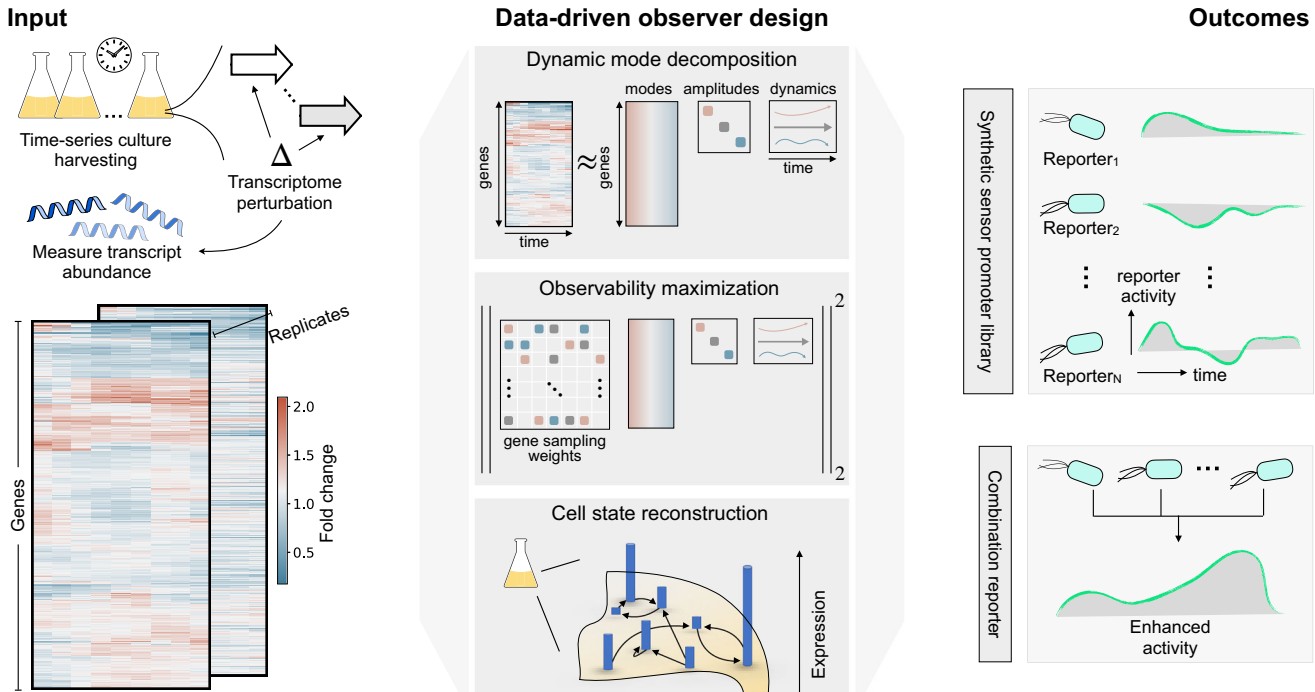

**Fig. 1 | Transcriptional genetic sensors underlying the response from envir-onmental perturbations can be extracted using data-driven sensor placement.** Bulk RNA sequencing (RNA-seq) measures transcript abundance over time fol-lowing transcriptome perturbations (left). Our method (middle) starts by applying dynamic mode decomposition (DMD) to the fold change response of the tran-scriptome of a population of cells to discover dynamic modes which govern the evolution of the cell state. The dynamic modes are used to design a state observer (or equivalently, optimal gene sampling weights) from which the contribution of each gene to the observability of the transcriptome dynamics can be assessed. The gene sampling weights provide an interpretable ranking for the selection of

*biomarkers* that are then used for cell state reconstruction. Our method returns: (1) a dynamics matrix (or equivalently, a set of dynamic modes) describing how expression of gene $i$ at time $t$ is impacted by gene $j$ and time $t − 1$. and (2) gene sampling weights signifying a gene's contribution to the observability of the cell state. The outcome (right), demonstrated in this work, is a library of synthetic analyte-responsive promoters (genetic reporters) that are used to detect an analyte of interest. Since each genetic reporter has a unique response to the same per-turbation, the library can be pooled at the assay level to fuse the reporter responses, resulting in enhanced analyte detection.

state, $\mathbf{z}_k$ for some time point $k$, we are concerned with for discovery of genetic reporters. Of the nearly 6000 known genes in the SBW25 genome, a large fraction of them were not expressed at significant levels. We filtered genes with TPM < 100 and specifically only 10% of or 624 genes are kept for modeling and analysis due to their relatively high abundance.

Given our goal of extracting salient analyte-responsive promoters from time-series gene expression data, we first model the dynamical process that is driven by the input of malathion on the SBW25 tran-scriptome. In the next section, we apply dynamic mode decomposition (DMD) to approximate the fold change response with a sparse col-lection of dynamic modes. We demonstrate how DMD can accurately describe gene expression dynamics by decomposing the time-series gene expression into temporally relevant patterns.

### Dynamic mode decomposition uncovers modes of host cell response

Dynamic mode decomposition (DMD) is a time-series dimensionality reduction algorithm that was developed in the fluid dynamics com-munity to extract coherent structures and reconstruct dynamical systems from high-dimensional data[35]. Recently, several works have adapted and applied DMD to biological systems in various contexts[49–53], choosing DMD for its ability to (i) reproduce dynamic data over traditionally static methods such as principal component[54] or independent component analysis[55] and (ii) represent the dynamics of high-dimensional processes, in our case gene interaction networks, using only a relatively small number of modes.

To uncover the diverse modes of the host cell response to mala-thion induction, we performed (exact) DMD[37] on the transcriptomic dataset (see "Methods" section for the details). Specifically, we per-form exact DMD on the standardized fold change, $\bar{\mathbf{z}}$, which decom-poses a gene expression matrix (genes × time points) into dynamic modes, eigenvalues, and amplitudes in the form

$$\hat{\mathbf{z}}_t = \sum_{i=1}^{r} \mathbf{v}_i \lambda_i^t b_i = \mathbf{V}\boldsymbol{\Lambda}^t\mathbf{b} = \mathbf{V}\boldsymbol{\Lambda}^t\mathbf{V}^{-1}\mathbf{z}_0 \qquad (1)$$

where the rank $r$ reconstruction of the cell state at time $t$ is $\hat{\mathbf{z}}_t$, $\mathbf{v}_i$ are the learned dynamic modes, $\lambda_i$, are the learned eigenvalues, and $b_i$ is the amplitude associated with each dynamic mode (often known as load-ing in the dimensionality reduction literature). From this we see that the transcriptome dynamics are modeled by a sum of damped, forced, and unforced sinusoidal behavior when the magnitude of the eigen-values are less than one, greater than one, or exactly equal to one, respectively. This decomposition constructs a low-dimensional linear model from high-dimensional time-series data; quantitative features of a nonlinear model are not captured in our model, e.g. multiple equilibria and chaos. If these nonlinear features are relevant to the system being studied, one can extend DMD to capture arbitrary nonlinearities, at the cost of requiring a larger number of samples to infer the parameters of the nonlinear function[56]. In this section we will describe how modeling the fold change response with DMD enables the identification of biologically relevant temporal patterns that are driven by the malathion perturbation.

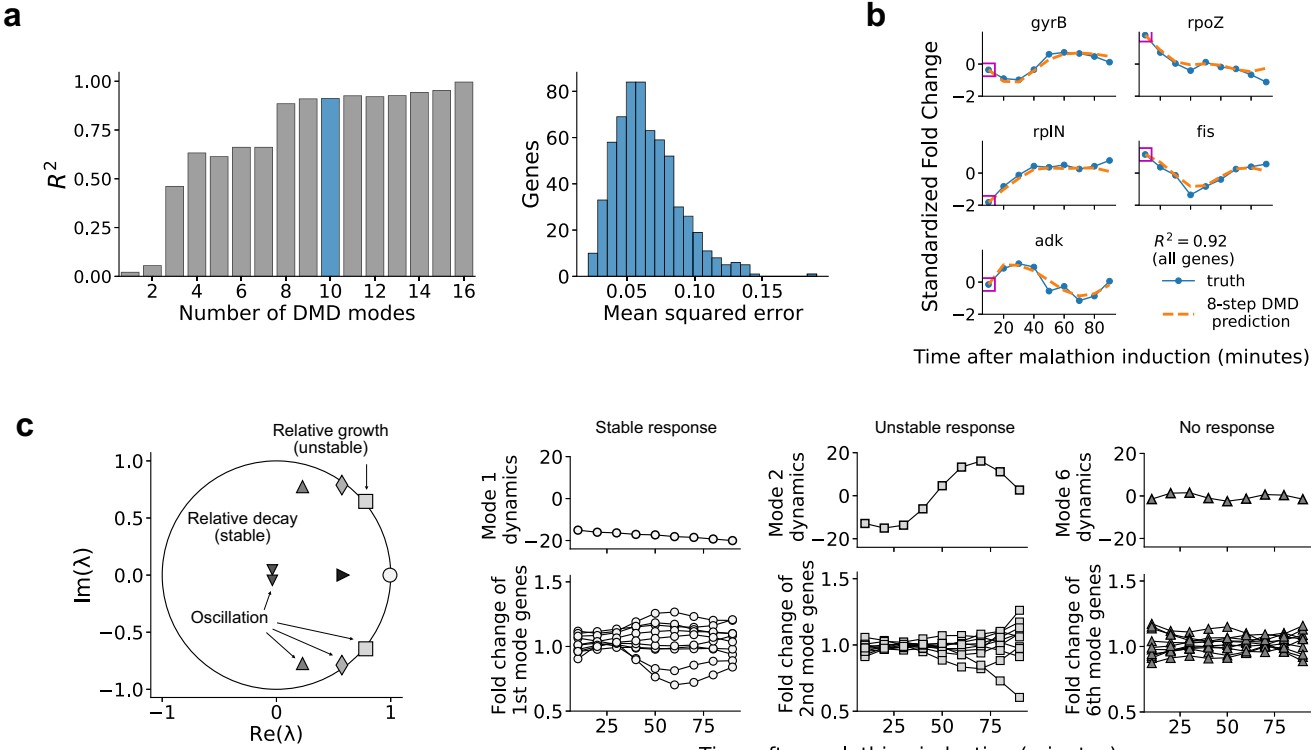

**Fig. 2 | Dynamic mode decomposition provides a predictive and interpretable model of gene expression dynamics. a** The coefficient of determination for the reconstruction is shown while varying number of DMD modes, $r$ in (1) (left). 10 DMD modes are used to construct transcriptome dynamics in this work and the mean-squared error per gene is shown in the histogram on the right. **b** The eight-step prediction is visualized for five randomly selected genes in the transcriptomic dataset. The mean fold change across the two biological replicates (blue solid curve) and across predictions (orange dashed curve) are depicted. Magenta squares overlapping each gene's initial condition indicates the data that is provided to make predictions. The coefficient of determination, $R^2$, for the eight-step

prediction across all genes is computed to be 0.92. **c** (Left) The DMD spectrum reveals the growth, decay, and oscillation of each of the 10 dynamic modes that comprise the transcriptomic dataset. Each marker is an eigenvalue, and its diameter is proportional to the magnitude of the corresponding dynamic mode. Eigenvalues inside the unit circle correspond to decaying dynamics, eigenvalues with nonzero imaginary part correspond to oscillatory dynamics, and eigenvalues outside the unit circle correspond to growing dynamics. (Right) The eigenvalue scaled amplitudes, $\lambda_i^t b_i$, of modes 1, 2, and 6 are visualized (upper) along with the 10 genes whose dynamics are most impacted by each of the modes (lower). The marker used for each mode indicates which eigenvalue it corresponds with in **c**.

We found that 10 dynamic modes provide an optimal balance between predictive accuracy and model instability. As the number of modes, $r$, is increased, we see monotonically increased predictive accuracy as measured by the coefficient of determination ($R^2$) (Fig. 2a (left)). However, the number of eigenvalues with magnitude greater than one, i.e. unstable modes, also increases with the number of modes (Supplementary Fig. 3). As we will discuss in further detail in the next section and in the Methods, instabilities introduce challenges in observability analysis, therefore we aimed to minimize the presence of unstable modes in the learned dynamics. Although, since predictive accuracy is important, we could not altogether remove unstable modes.

Using the 10 dynamic modes, we obtain an accuracy of 0.92 as measured across all genes. Figure 2b shows a set of 5 genes and their temporal predictions using the DMD model. The predictions are computed by feeding an initial condition (the gene expression at time $t = 0$) to the model and then predicting the expression at all all subsequent time points; the nine time points in the dataset. This amounts to two eight-step predictions across the biological replicates. We emphasize that this is distinct from measuring model accuracy by computing a one-step prediction for each time point, which gives very little information about the dynamic process that has been captured. The low-dimensional model learned via DMD has accurately captured the dynamics of the fold change response. To provide a foundation for understanding when linear models can accurately represent fold change dynamics, we have shown, in the Supplementary Information

(Section 1.4), that the fold change response of two linear systems, under stated assumptions, can be represented as the solution of a linear system.

Our DMD analysis uncovers three distinct modal responses of the malathion-perturbed transcriptome dynamics, namely stable, oscillatory, and unstable responses. We classify each mode's response type by the behavior of the associated eigenvalue. If the associated eigenvalue has magnitude <1 or >1, the mode is classified as stable and unstable, respectively. If the eigenvalue also has a nonzero imaginary part, the mode is classified as oscillatory as well. We have plotted the 10 DMD eigenvalues relative to the unit circle in Fig. 2c and labeled the eigenvalues according to their type.

Stable modes are characterized by eigenvalues which are inside the unit circle. The magnitude of eigenvalues inside the unit circle are strictly less than one and such a set of stable modes indicate relative decay, that is to say that many genes have a temporal response which only transiently deviate from a neutral fold change (fold change equal to one for non-standardized trajectories and fold change equal to zero for standardized trajectories). Stable modes that have eigenvalues nearer to the unit circle are capturing majorly uninhibited genes, while stable modes that are nearer to the origin are capturing genes which converge to neutral fold change exponentially, i.e. they exhibit strong relative decay in their fold change.

Dynamic modes which are oscillatory are characterized by by eigenvalues with nonzero imaginary part. Since gene expression data is

always real-valued, oscillatory modes will always come in complex conjugate pairs. Each pair of complex-valued modes then describes a fixed frequency of oscillation, and each gene's dynamics can be reconstructed from one or more of these frequencies. The work of Sirovich found that the oscillatory modes obtained from DMD represent the genes underlying the yeast cell cycle, and the frequencies of oscillation were shown to provide an estimate of the cell cycle period that agrees with the literature[51].

Unstable modes are characterized by eigenvalues whose magnitude is larger than one. Many genes show temporal response that were either upregulated or downregulated. If the upregulation and downregulation is persistent throughout the gene's temporal profile or occurs at later times, there must be at least a single mode with eigenvalue outside the unit circle to be able to capture the underlying unstable response. This is because DMD is essentially learning a linear state-space representation of the fold change response and a linear system can only exhibit three types of limiting behaviors, (i) convergence to the origin (stable), (ii) periodic orbits, and (iii) divergence to infinity (unstable). Therefore, for the reconstruction accuracy to be maximized, DMD eigenvalues with magnitude larger than one may be necessary. Such eigenvalues are marked with relative growth in Fig. 2c. Though the two unstable eigenvalues are outside the unit circle, they are only marginally so, implying that unstable trajectories make up only a small portion of the transcriptomic response to malathion.

Despite the fact that most genes require a superposition of all of the dynamic modes for accurate reconstruction, we show that the modes can successfully group genes into interpretable clusters. Figure 2d (upper) shows the evolution of three dynamic modes ($\lambda_i^t b_i$) representative of the transcriptomic dataset: modes 1, 2, and 6, corresponding to stable (modes 1 and 6) and unstable (mode 2) directions in gene space. The loading of mode $j$ on gene $i$, $\mathbf{V}_{ij}$, can be used to identify genes which are most influenced by the corresponding mode. In this way, we can use the DMD modes to cluster temporal responses in gene space, providing an interpretation to each DMD mode. The temporal gene clusters are shown in Fig. 2d (lower).

The genes which are most influenced by mode 1 are those which diverge, in a stable manner, from a neutral fold change while the genes most influenced by mode 2 are those which diverge away from neutral fold change, capturing unstable trajectories. This is consistent with the eigenvalues of mode 1 and mode 2, which are stable and unstable, respectively. Finally, the genes most influenced by mode 6 are those with no clear trend present in their dynamics. In the next section, we will characterize those genes which contribute to cell state reconstruction and act as reporters for the malathion specific response. Relatedly, of the 20 genes that are most impacted by mode 1, seven of these genes contribute highly to cell state reconstruction (they are within the top 20 genes that contribute to the observability of the system).

The results of this section demonstrate that the set of 10 recovered DMD modes, eigenvalues, and amplitudes are indeed biologically relevant to the dynamics of the malathion response in the window of time that we have sampled the transcriptome. A key takeaway is that gene expression dynamics sampled at the resolution of minutes can be well approximated by a linear dynamical system, i.e. by a set of exponentially shrinking and growing modes. In what follows, we develop a sensor placement framework, relying on the learned linear dynamical system, to generate a ranked list of biomarker genes, i.e. subsets of genes which show variation to malathion induction and that can recapitulate the cell state.

## Sensor placement for cell state inference and extraction of genetic sensors

Gene interaction networks are complex systems that induce systematic interdependencies between genes. That is to say that the expression of most genes, if not all, depends on the expression of at least one more genes in the network. These interdependencies make it possible to measure only a subset of genes to infer the behavior of all other genes[57]. The approach taken in this work for evaluating whether a gene is an encoder of cell state information is to quantify how much each gene contributes to observability. To do this, we optimize a scalar measure of the observability gramian, a matrix which determines the amount of information that a set of sensors can encode about a system. Specifically, if we let the DMD reconstruction of the cell state be rewritten as $\hat{\mathbf{z}}_t = \mathbf{V}\mathbf{\Lambda}\mathbf{V}^{-1}\mathbf{z}_{t-1} = \mathbf{K}\mathbf{z}_{t-1}$ and define an output equation

$$y_t = \mathbf{w}^\top \hat{\mathbf{z}}_t \tag{2}$$

where $\mathbf{w}$ is a vector of weights, called sampling weights, that define the contribution of each gene to the output of the system, then we define the observability gramian[58] as

$$\mathcal{X}_o = \sum_{i=0}^{\infty} \mathbf{K}^{i^\top} \mathbf{w}\mathbf{w}^\top \mathbf{K}^i. \tag{3}$$

In the context of transcriptome dynamics, given the DMD representation of the dynamics, $\mathbf{K}$, and a chosen gene sensor placement, $\mathbf{w}$, the gramian quantitatively describes (i) to what degree cell states are observable and (ii) which cell states cannot be observed at all. Increasing (i) while decreasing (ii) is the aim of many sensor placement techniques; furthermore, many scalar measures of the gramian have been proposed to determine the sensor placement (the weights $\mathbf{w}$) which maximize the observability of the underlying dynamical system[59–61]. Many of the proposed approaches require explicit computation of the observability gramian, which can be computationally expensive for high-dimensional networks and intractable for unstable systems.

Here we develop an optimization framework which does not require explicit computation of the gramian. We do this by maximizing the *signal energy*, $\sum_{i=0}^{T} y_i^2$, of the underlying system. The resulting sensor placement problem is then defined to be an integer program in which the weights can only takes binary values 0 or 1. As high-dimensional integer programs are known to be computationally intractable, we employ several relaxations on the problem. The details of the full sensor placement problem and the relaxations are presented in the Supplementary Information (Section 1.2). Notably, we have approximated the full sensor placement problem to one in which an analytical solution always exists. This reduces the overall computational complexity, providing an approach which scales for a wide array of high-dimensional biological datasets collected from diverse host organisms.

The strategy we employ is to assign gene sampling weights, $w_g$, to each gene $g$ through optimizing sensor placement, i.e., maximizing the signal energy. The significance of the magnitude of each weight is to rank each gene by their contribution to observability. The Methods section provides quantitative details on the relationship between observability, the observability gramian, and signal energy for sensor placement. In the Supplementary Information (Section 1.3), we provide a brief exposition of the sensor placmenet problem on simulated systems. We show how the sampling weights are affected by network topology and the number of time points.

By examining the learned gene sampling weights, we found that nearly all 624 modeled genes contribute, many insignificantly, to the observability of the system. Displayed in Fig. 3a (upper) are the magnitude of gene sampling weights, $\mathbf{w}$, normalized by the standard deviation of the corresponding gene, that maximize the observability of the cell state. Weights that are negative-valued (only magnitudes are shown here) correspond to downregulated genes and weights that are positive-valued correspond to genes that are upregulated. The higher the magnitude of the gene sampling weight, the more important the gene is likely to be for cell state reconstruction. The lower portion of

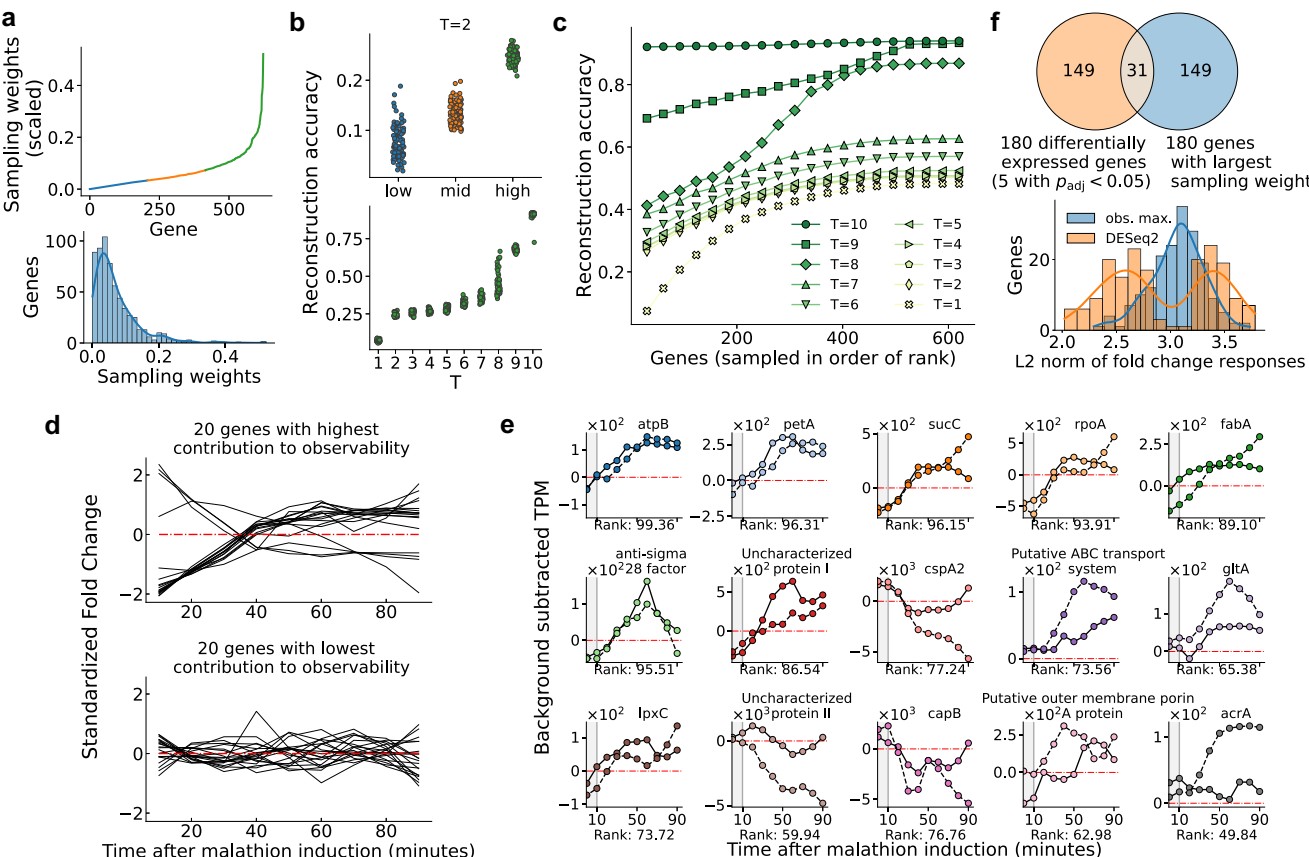

**Fig. 3 | Gene sampling weights which maximize observability provide machine learned ranking for extraction of genetic sensing elements. a** The gene sampling weights, $\mathbf{w}$, normalized by standard deviation of the corresponding gene, sorted by magnitude and plotted in the upper panel. The weights are grouped into three categories: (i) the third of genes with highest magnitude of sampling weights (green), (ii) the third of genes with second highest magnitude of sampling weights (orange), and ii) the lower third that remains (blue). The lower panel is a histogram of the sampling weights and a kernel density estimate is superimposed. **b** The reconstruction accuracy ($R^2$) between the true initial condition and the estimated initial condition when sampling 50 genes at random from each of the aforementioned groups for $T = 2$ time points (top). (Bottom) The reconstruction accuracy for the high group as a function of $T$. **c** Reconstruction accuracy between the estimated initial condition $\hat{\mathbf{z}}_0$ and the actual $\bar{\mathbf{z}}_0$ is plotted for number of sampled time points $T = 1$ to $T = 10$. **d** The average fold change response of each of the 20 genes which contribute most (top) and least (bottom) to the observability of the initial cell state are plotted. **e** The background subtracted TPM (malathion (TPM) − negative control (TPM)) of the 15 biomarker genes selected from the proposed ranking−by contribution to observability. The label on each x-axis indicates the percentage rank (out of 624 genes) of the gene, with respect to the gene sampling weights, with 100% corresponding to highest rank. The two biological replicates are shown using solid and dashed lines, respectively. Malathion was introduced to the cultures after collecting the sample at 0 minutes, hence this sample is not used for modeling and cell state inference (shaded in gray). **f** A Venn diagram comparing 180 differentially expressed genes and genes with the largest sampling weights identified by our approach (top). DESeq2[16] was used to identify differentially expressed genes. The bottom panel shows a histogram of the L2 norm (Euclidean distance from the origin) of the fold change responses for the genes in the unique sets in the Venn diagram.

Fig. 3a shows the histogram of the sampling weights in $\mathbf{w}$, displaying that there are fewer higher magnitude genes overall. To test the notion that genes with higher weights contribute more to the observability, the sampling weights are artificially grouped into three categories, distinguishing genes which correspond to the top (green), middle (orange), and lower (blue) third for magnitude of sampling weights. Each category contains 208 genes, and next we show the gain in information that can be achieved when sampling from one category over another.

To examine the contribution to observability provided by genes in each of the categories, we perform Monte Carlo simulations to estimate the expected predictability of the initial cell state. From output measurements, $y_t$ ($t = 1, 2, \ldots T$), that are generated by randomly sampling 50 genes from a specified category (low, mid, high), the initial cell state, $\bar{\mathbf{z}}_0$, is estimated and the coefficient of determination ($R^2$) between the actual and estimated cell state is computed as a measure of reconstruction accuracy. The simulation is repeated 100 times for each category and the resulting distributions over the random gene sets are plotted in Fig. 3b. In the top panel, we can see that when $T = 2$ (2

time points are used for reconstruction), predictability of the cell state is highest for the genes in the high category. Specifically, the reconstruction accuracy is three and two times larger in the high category than in the low and mid categories, respectively. In the lower panel we show how the reconstruction accuracy changes with changing the number of time points, $T$, for the high group of genes. We find that that reconstruction accuracy monotonically increases with $T$, however we point out that due to not being able to accurately capture network topology from sparse data, the results gathered at large $T$ ($T \geq 6$) do not show significant differences between the groups. This is due to the fully-connected topology of the state-space model we have learned using DMD. The interdependencies between genes (though mostly miniscule) are amplified exponentially over time, resulting in highly observable genes transferring information to lowly observable genes. Hence, it may not be possible to distinguish reconstruction accuracy of the groups of genes when evaluated at large times.

Measuring fewer genes for many time points leads to higher cell state reconstruction accuracy than if many genes are measured for fewer time points. This result is demonstrated in Fig. 3c which shows

how the cell state reconstruction accuracy is affected by two parameters, the number of sampled genes and the number of time points, $T$, that the genes are measured for. The reconstruction accuracy is again the coefficient of determination, $R^2$, between the reconstructed initial condition, $\hat{\mathbf{z}}_0$, and the actual initial condition $\bar{\mathbf{z}}_0$. For each $T$, the first data point is generated by sampling only the five genes with the highest sampling weights for $T$ time points. The complete cell state is then inferred from these measurements alone and the coefficient of determination between the estimated and actual cell state can be computed (see "Methods" section for a detailed description of the cell state inference algorithm). To compute subsequent data points, the next five genes with maximum sampling weights are simultaneously measured along with previously measured genes, and the cell state is reconstructed again. For the response of SBW25 to malathion, we find that even if only the top five genes are measured but for $T = 10$ time points, the cell state reconstruction is still more accurate than if all genes with nonzero sampling weights are measured with $T \leq 8$ time points. Specifically, the reconstruction accuracy with 5 genes sampled for $T = 10$ time points is nearly 0.9 while the reconstruction accuracy with 600 genes sampled for $T = 8$ time points is slightly >0.8. This signifies that the ability to study the dynamics of a few genes with fine temporal resolution can greatly increase the knowledge of the entire system.

Failure to reconstruct the initial cell state is a result of two mechanisms. The first is that we only have access to the DMD representation of the dynamics, not the true dynamics. Therefore, any output measurements generated using the DMD model will certainly incur an error with respect to the actual dynamics. As error accumulates each time-step, it is possible for the reconstruction accuracy to decrease with increasing time points. The second hindrance for full cell state reconstruction is when many genes contain redundant information. If two genes have nearly identical gene expression profiles, adding the second gene to the set of measurements provides no useful information for the cell state inference. This may explain the asymptotic behavior of the curves in Fig. 3c. There are only relatively few distinct dynamic profiles present in the transcriptomic dataset, and once all distinct profiles have been sampled, no further improvement in reconstruction can occur. This explanation is consistent with the fact that many genes co-express[21] and this fact has even been used to reconstruct dynamic gene regulatory networks[62].

The gene sampling weights, $\mathbf{w}$, provide a machine learned ranking for discovering genetic biomarkers. Recall that the fold change was taken to be the state of the system when performing DMD. In so doing, we show that the observability-based ranking can also predict genes that respond to malathion in a condition specific manner. Specifically, genes which contribute highly to the observability of the system are genes which show prolonged dysregulation in the presence of malathion. This is visualized in Fig. 3d where in the top panel the 20 genes which have the largest sampling weights are plotted. Each of the 20 genes show dysregulation from the neutral fold change (0) that is persistent over the course of the time-series. Conversely, the 20 genes with lowest sampling weights show no clear trend or signal of dysregulation. Significant correlations are present among the genes which contribute highly to observability. This is due to the fact that we have solved a relaxed version of the sensor placement problem that allows each gene to have nonzero weight towards maximizing the observability. In the unrelaxed problem, only a pre-defined number of genes can have nonzero weight and therefore to capture all the distinct temporal profiles in the transcriptomic dataset, selected genes are likely to be uncorrelated.

To show that observability-ranked genes can act as genetic reporters for malathion, we selected a set of 15 genes with which to construct transcriptional reporters from. We considered several approaches for gene selection that depend on the learned weights in Fig. 3a and the criteria we considered were gene rank, correlations among the chosen gene set, and/or the ability of the gene set to recapitulate the cell state. Correlations among the gene set are considered because they can be used as a predictor for the variety of responses the library will generate after being subjected to malathion.

By choosing the top ranked genes we can sample top malathion responders, though as can be seen in Fig. 3d they are highly correlated with each other. The correlations between the top responders are shown in Supplementary Fig. 5b. The variety of reporter responses from this library of genes is predicted to be low, however the biological processes that the top responders are involved in is diverse (Supplementary Fig. 5c).

Since high variety of reporter response is of interest, we further consider a correlation-based strategy for gene selection which first selects the top ranked gene, then subsequently selects genes from the ranking only if its correlation with previously selected genes is less than a chosen threshold. In Supplementary Fig. 6, the RNA-seq temporal profiles of genes selected using a correlation threshold of 0.5 are visualized. To measure the gene set correlation and compare across gene sets, we introduce the following metric,

$$C = ||\mathbf{1}_{k \times k} - \mathbf{R}^{abs}||_F \tag{4}$$

where $\mathbf{1}_{k \times k}$ is the matrix of $k \times k$ ones and $\mathbf{R}^{abs}$ is the element-wise absolute value Pearson correlation coefficient matrix of the $k$ selected genes. When the metric approaches zero, the overall correlation between the selected gene set is large. Conversely, when the metric approaches infinity, the overall correlation between the selected gene set is small. For the gene set chosen using the correlation-based strategy, $C = 7.8$, while for the gene set chosen using only the ranking, $C = 2.9$.

Next, we want to consider the cell state reconstruction ability of gene sets. If we evaluate the reconstruction ability for $T = 8$ time points using the top responders, we obtain an accuracy of 0.39, while for the correlation-based selected genes, we obtain an accuracy of 0.37. To obtain gene sets which have higher reconstruction accuracy, we employed a randomized, Monte Carlo approach for sampling gene sets. Isolating to the top half of the ranking, we selected 15 genes at random and tested their ability to reconstruct the cell state. In result, we obtained a gene set with reconstruction accuracy in the top 92% of tested gene sets ($R^2 = 0.51$; Supplementary Fig. 7). This set also has the lowest correlation metric of all tested gene sets in the top 15% of ranked genes (94 genes) as well as reconstruction accuracy in the top 95 percentile (Supplementary Fig. 7a). Moreover, the correlation metric for this gene set is $C = 7.0$, slightly below that of the gene set chosen using the correlation-based strategy. This gene set was then selected for the malathion reporter library as it has both high cell state reconstruction and low overall correlation.

The 15 time-series profiles for the selected genes generated via RNA-seq are visualized in Fig. 3e in the form of $\text{TPM}_{malathion} - \text{TPM}_{control}$. Of the 15 selected biomarker genes, 12 appear to be activated by induction of malathion while the remaining 3 appear to be repressed. Table 1 lists the molecular functions of each of the selected genes based on their Gene Ontology (GO) annotations[63]. Where gene names are not available, we have used protein annotations to denote those genes. It is shown that the set of molecular functions are diverse, indicating that malathion drives the activation and repression of disparate biological processes. When synthesized into genetic reporters, as we will show in the next section, these biomarker genes exhibit distinct dynamic range, sensitivity, and time-scales in response to malathion.

Comparing our approach to differential expression analysis, we find that our results are largely in complement to each other. To start, we used DESeq2[16] and found five significantly differentially expressed genes after multiple-testing correction with the Benjamini–Hochberg procedure. The fold changes of the five genes lie in the range 0.52–1.54

**Table 1 | Sensor promoter library metadata and transfer curve parameters for the fitted Hill equations in Fig. 4d**

| Malathion reporter | Locus tag | Molecular function | Act./Rep. | $y_{min}$ | $y_{max}$ | $K_M$ | $n$ |
|---|---|---|---|---|---|---|---|
| atpB | PFLU_6124 | Proton-transporting ATP synthase activity, rotational mechanism | Activated | 1467 | 1783 | 0.6 | 4.5 |
| petA | PFLU_0841 | 2 iron, 2 sulfur cluster binding, | Activated | 853 | 1125 | 1.4 | 2.4 |
| | – | Metal ion binding | – | – | – | – | – |
| | – | Ubiquinol-cytochrome-c reductase activity | – | – | – | – | – |
| sucC | PFLU_1823 | ATP binding | Activated | 257 | 337 | 0.2 | 1.9 |
| | – | Magnesium ion binding | – | – | – | – | – |
| | – | Succinate-CoA ligase activity | – | – | – | – | – |
| rpoA | PFLU_5502 | DNA binding | Activated | 1256 | 1542 | 0.9 | 3.0 |
| | – | Protein dimerization activity | – | – | – | – | – |
| | – | DNA-directed 5'-3' RNA polymerase activity | – | – | – | – | – |
| fabA | PFLU_1836 | Dehydratase activity | Activated | 292 | 373 | 0.2 | 1.1 |
| | – | Isomerase activity | | | | | |
| anti-sigma 28 factor | PFLU_4736 | Negative regulator of flagellin synthesis | Activated | 339 | 535 | 0.7 | 1.5 |
| Uncharacterized protein I | PFLU_3761 | – | Activated | 2465 | 3110 | 0.5 | 2.7 |
| cspA2 | PFLU_4150 | Major cold shock protein | Activated | 706 | 1186 | 1.5 | 5.3 |
| Putative ABC transport protein | PFLU_0376 | Ligand-gated ion channel activity | Activated | 584 | 1083 | 1.0 | 2.0 |
| gltA | PFLU_1815 | Citrate (Si)-synthase activity | Activated | 238 | 458 | 0.9 | 1.9 |
| lpxC | PFLU_0953 | Metal ion binding | Activated | 1017 | 2418 | 0.4 | 7.4 |
| | – | Deacetylase activity | – | – | – | – | – |
| Uncharacterized protein II | PFLU_1358 | – | Repressed | 1073 | 3387 | 0.3 | 1.9 |
| capB | PFLU_1302A | Cold shock protein | Repressed | 9616 | 10543 | 0.9 | 2.9 |
| Putative outer membrane porin A protein | PFLU_4612 | Porin activity | Activated | 642 | 1172 | 0.6 | 1.5 |
| acrA | PFLU_1380 | Transmembrane transporter activity | Activated | 354 | 682 | 0.9 | 2.9 |

and the control subtracted temporal responses are visualized in Supplementary Fig. 12. Of these five genes, our observability approach ranks two genes in the top 82% and four in the top 55% of observable genes.

To be able to make a larger comparison between the methods, we next used non-corrected p-values and a significance threshold of 0.05 to call a gene differentially expressed. Including the five genes that were significantly differentially expressed, this identified a total of 180 differentially expressed genes (however 175 of these genes cannot be called significant) after induction of malathion. Comparing these genes to the genes with 180 largest sampling weights, we find that there are 31 genes are in common (Fig. 3f, upper). To show the distinction between the genes identified between the two approaches, we visualize the histogram of the L2 norm of each gene's fold change response (Fig. 3f, lower). We see that our approach, labeled, obs. max., identified genes with fold change response centered around 3.0, while DESeq2 identifies two clusters of genes centered around 3.2 and 2.5.

Overall, differential expression analysis is not always suitable for a dataset with low number of biological replicates and can result in very few genes being called as differentially expressed. After multiple-testing correction, 5 out of 6009 genes were called significantly differentially expressed, and there is considerable overlap between the observability ranking and these genes. The results may indicate that the two methods may converge in the case of large number of biological replicates. However, in the case of low biological replicates, our approach, as we show in the next section, identifies numerous malathion responsive genes that differential expression analysis was not able to identify. This indicates that our observability framework is useful for analysis of time-course RNA-seq measurements.

## Design and characterization of fluorescent malathion sensors
To validate the transcriptome-wide analysis for identification of analyte-responsive promoters, the putative promoters of the

candidate sensor genes were cloned into a reporter plasmid containing a reporter gene encoding *sfGFP* (superfolder green fluorescent protein) and transformed into the host SBW25 (Fig. 4a). The reporter strains are cloned in an unpooled format, allowing for malathion response curves to be generated at the reporter level as opposed to a pooled study which would incur additional sequencing costs for individual strain isolation.

Malathion reporters are characterized in the laboratory in an environmentally relevant way by sourcing malathion from the commonly used commercial insecticide called Spectracide (containing 50% malathion). First, it was verified that the response of the reporters to analytical standard malathion was consistent with the response when induced with Spectracide. That is to say that if the reporter was upregulated (downregulated) in response to malathion, it was also upregulated (downregulated) in response to Spectracide. Furthermore, the culture media containing nutrients and Spectracide that the reporter strains were cultured in was analyzed with mass spectrometry and compared to the mass spectrum of analytical standard malathion. Comparing the two mass spectra, we found that they are nearly identical (Supplementary Figs. 14–26). See the "Methods" section for more details about the use of Spectracide as a source for malathion and Supplementary Fig. 9 for the effect of Spectracide on the growth of the reporter strains.

To examine the transcriptional activity of *sfGFP*, controlled by the biomarker gene promoters, cells are grown in rich medium and fluorescence output was measured every three minutes over 24 hours of growth. This resulted in 400 time points per reporter strain, a nearly 45 fold increase over the number of time points obtained via RNA-seq see Supplementary Fig. 10. Prior to starting the experiment and collecting fluorescence measurements, reporter strains were induced with Spectracide to drive the reporter response. Due to the long half-life and fast maturation time of *sfGFP*[64], the reporter protein can accumulate inside the cell and does not accurately represent the mRNA abundance – which is subject to fast degradation by

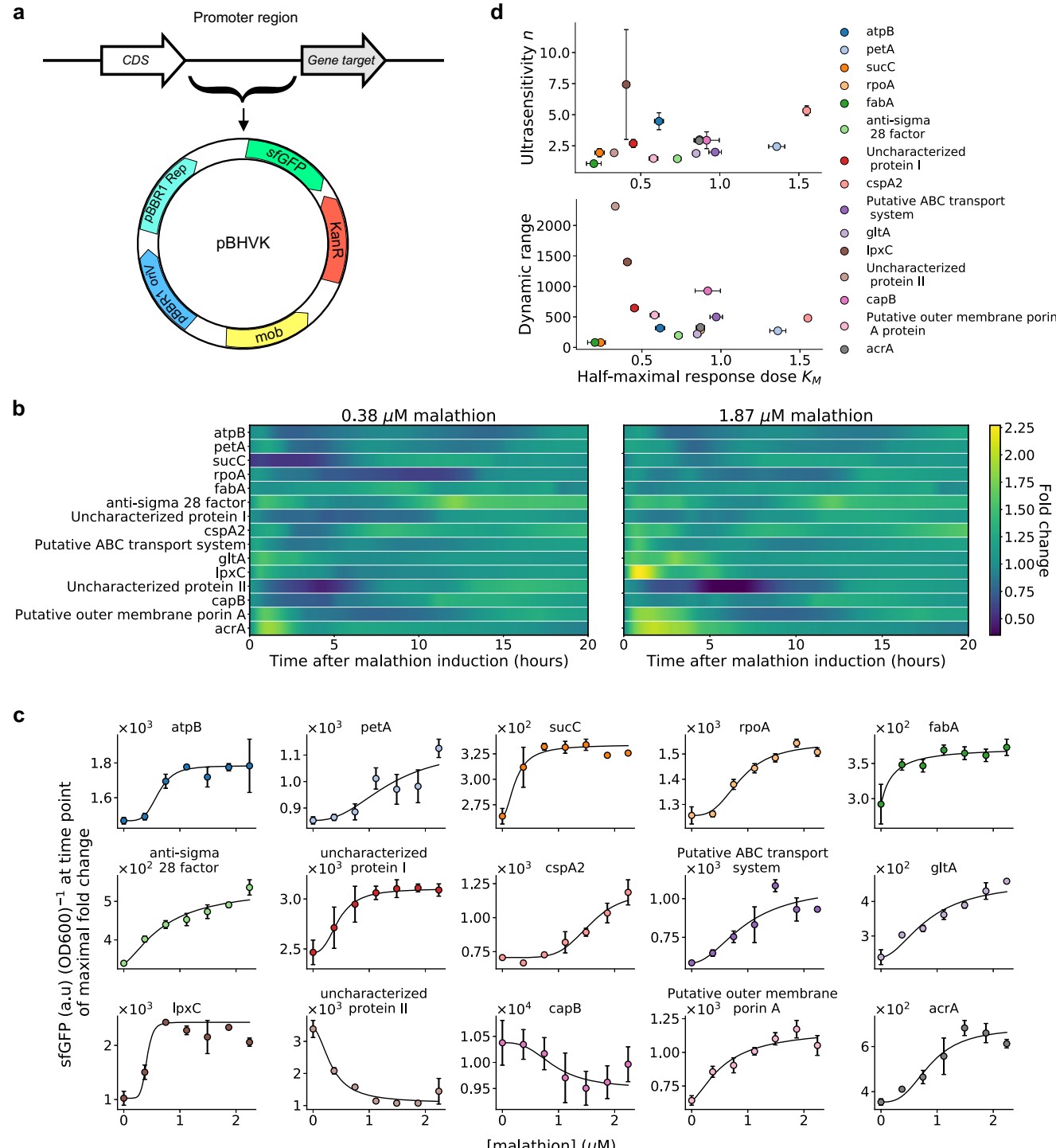

**Fig. 4 | Our machine learning approach successfully extracted 15 sensors, each with distinct malathion response curves. a** A map of the plasmid, pBHVK, used to construct the library. The plasmid contains a kanamycin resistance gene as well as a fast-folding *sfGFP* gene. **b** Average per cell *sfGFP* signal at 0.37 μM (left) and 1.83 μM (right) malathion normalized by signal at 0.0 μM malathion is shown for all 15 engineered strains. **c** Transfer curves (or dose-response curves) for each strain are depicted with markers and their fit to Hill equation kinetics are given by solid lines. The Hill equation parameters are given in Table 1. The promoter sequences corresponding to each reporter and time points for each transfer curve are given in Supplementary Tables 2 and 4, respectively. The transfer curves are plotted at the time point of maximal fold change of the 2.24 μM response with respect to the 0 μM response. The error bars represent the standard deviation from the mean across three biological replicates. **d** Transfer curve parameters for the dose-responses depicted in **c**. The error bars represent the standard deviation from the mean across three biological replicates. Source data are provided as a Source Data file.

ribonucleases. This results in the genetic reporters serving as a proxy for the rate of transcription initiation over time, rather than mRNA abundance. This is distinctly different from the transcript abundance that is measured via RNA-seq due to the instability of mRNA molecules.

Examining the transcription initiation driven by malathion at distinct concentrations reveals detailed gene expression dynamics, dependencies of expression on malathion concentration, as well as the correlations. Firstly, the fold change (with respect to 0.0 μM malathion

and referred to as the background) reveals oscillatory signals in several strains; the reporters *atpB*, *petA*, *cspA2*, and *acrA* each contain oscillations that are near in phase at 0.38 μM malathion (Fig. 4c). As the concentration of malathion is increased, only *atpB* and *petA* appear to remain in phase while the signals of the other strains strongly increase. We also see that *anti-sigma 28 factor* and *rpoA* oscillate with lower frequency and that *anti-sigma 28 factor* hits a peak around 10 hours after induction while *rpoA* hits an anti-peak around 10 hours after induction. For the lower malathion concentration, *sucC* has a large lag time until transcriptional activation occurs, however there is a sharp decrease in the lag time at the higher concentration. The strains *acrA*, *gltA*, *putative outer membrane porin A*, *putative ABC transport system*, and *lpxC* consistently respond within minutes of malathion induction with *lpxC* being the reporter with highest signal over background and *acrA* the reporter with highest overall signal energy (area under the curve) in early times. Though *cspA2* was shown by the RNA-seq data to be repressed by malathion, we find that *cspA2* strain is consistently activated in the presence of malathion. Of the remaining repressed promoters, *uncharacterized protein II* is far more repressed in the presence of malathion across all concentrations tested.

The response curves of the reporter strains to malathion can be mathematically characterized by Hill functions[65] ("Methods" section) which are described by two parameters. The first parameter is the Hill coefficient or cooperativity, *n*, which is a measure of how steep the response curve is. This is also denoted as a measure of ultrasensitivity which results in sigmoidal like response curves. The second parameter, $K_M$, is the Michaelis constant and it is equal to the malathion concentration at which the response is half of its minimum value subtracted from its maximum value. Figure 4d shows the malathion response curves of each reporter strain at the time point with maximum fold change with respect to the 0 μM malathion condition. The solid line depicts the fit of a Hill function to the experimentally generated response curves and the parameters of each Hill function are given in Table 1. The response shown is the average fluorescence per cell obtained by normalizing the sfGFP signal by the optical density. See Supplementary Table 4 for the precise time points used here for each strain and see Methods for further details on parameter fitting.

We find that there is significant variation across the Hill coefficient, dynamic range, and Michaelis constant in the library of reporters (Fig. 4d). The Hill coefficient, *n*, ranges from 1.1 to 7.4, and recalling that this parameter is a measure of sensitivity, the extremes depicted by a small slope in strain *fabA* and large slope in strain *sucC*, respectively. The dynamic range, measured as the difference between the maximum signal and the minimum signal, ranges from 80 to 1401 and is obtained by *sucC* and the repressed *uncharacterized protein II*, respectively. The Michaelis constant ranges from 0.2 to 1.5, depicted by the shift in malathion concentration at which half of the maximum signal is achieved from *fabA* and *cspA2*.

Overall, we find that each synthetic reporter, selected via our data-driven sensor placement framework, is capable of detecting malathion with distinct dynamic ranges and sensitivity. We next sought to characterize the specificity of the reporters to malathion. We note that two of the selected reporters, *ABC transporter* and *acrA*, are membrane transporters which often respond to many environmental stimuli.

Through screening of our reporter library with four other compounds, we found in comparison that the response of the reporters to malathion is unique. To characterize the specificity of reporting to other pesticides, we tested with zeta-cypermethrin and permethrin, two frequently used pesticides. To test whether the reporters response changes due to overall changes in metabolism, we tested with the two sugars fructose and lactose. The concentration of the pesticides were 1.87 μM to be consistent with previous malathion screens and the concentration of fructose and lactose were 14.2 and 7.5 μM, respectively. The time-lapse response of all 15 reporters to the four compounds and malathion are shown in Supplementary Fig. 11.

In Fig. 5a we show the Pearson correlation coefficients between reporter responses to malathion and reporter responses to the four other compounds. The rows of the heatmap show how correlated the malathion response of a single reporter is across compounds while the columns show the overall correlation of a compound response to the malathion response for the 15 reporters. The correlation metric shows that induction with permethrin is most (linearly) related to malathion response while induction with zeta-cypermethrin is least related to malathion response.

Though the correlation coefficient between malathion response and other compounds may be high for several (reporter, compound) combinations (e.g. *lpxC* and *acrA*), the time-lapse response of the reporters show significant deviations across comopund in their transient response. The top row of Fig. 5b show the fold change response of *lpxC* and *acrA* after perturbation with each of the compounds. We see that at early times, the response due to malathion is significantly larger compared to the response due to other compounds. At later times, each of the responses converges to a neutral fold change, resulting in an overall high correlation. The bottom row Fig. 5b shows the fold change response of two reporters with overall negative correlation across compounds. Here we see that both the transient and long-term responses of other compounds deviate from the malathion response. Of the 15 reporters, 10 of them show no significant response to fructose or lactose (see Supplementary Fig. 11), indicating that overall the selected genes are not responding to broad changes in metabolism.

Overall, through observability analysis for extraction of sensor promoters and through the analysis presented in Fig. 5a, we find that the set of 15 reporters acts as a 15-dimensional cell state that can be used for malathion sensing and detection. Though we cannot conclude from our experiments and analysis that malathion directly interacts with any single promoter we have extracted, the 15-dimensional fingerprint provided by our reporters has been shown to be reproducible and unique among the tested compounds. Our approach to detection has limitations; since we cannot compare the response to malathion with the response to an exhaustive list of compounds, it may be the case that molecules which are highly similar in structure to malathion may induce similar responses.

## Pooling reporters at the assay level results in an enhanced malathion reporter

One criteria we optimized for in the malathion reporter library was diversity of response. As opposed to a set of reporters with similar responses, when aggregating diverse responses a unique output can be expected. Thus, we next established an experimental assay for pooling of individual reporters; this is in lieu of constructing a combinatorial promoter which can be challenging due to the curse of dimensionality when combining *n* promoters out of a set of *N* total promoters.

The motivation for such an experiment is due to the difficulty of strain isolation in an environmental setting. In order for our library of reporters to prove useful in the field, they should be able to operate in tandem without negative effects on the malathion response. To measure the response of all reporters in a pooled fashion, we first cultured all 15 reporter strains individually. Then before taking measurements, we pooled all the strains in equal numbers by carefully measuring the density of each culture. The pooled culture was then put into a plate reader and the *sfGFP* and optical density were measured over time.

Pooling all 15 genetic reporters results in a salient malathion response. The time-lapse curves of the *sfGFP* normalized by cell density are shown in Fig. 6a and the fold change 24 hours after malathion induction is shown in Fig. 6b for varying concentrations. At a malathion induction concentration of 1.12 μM, the pooled reporter exhibits a sustained response after an initial transience with a fold change of 2.1 after 24 h of growth. In contrast, the maximum fold change achieved

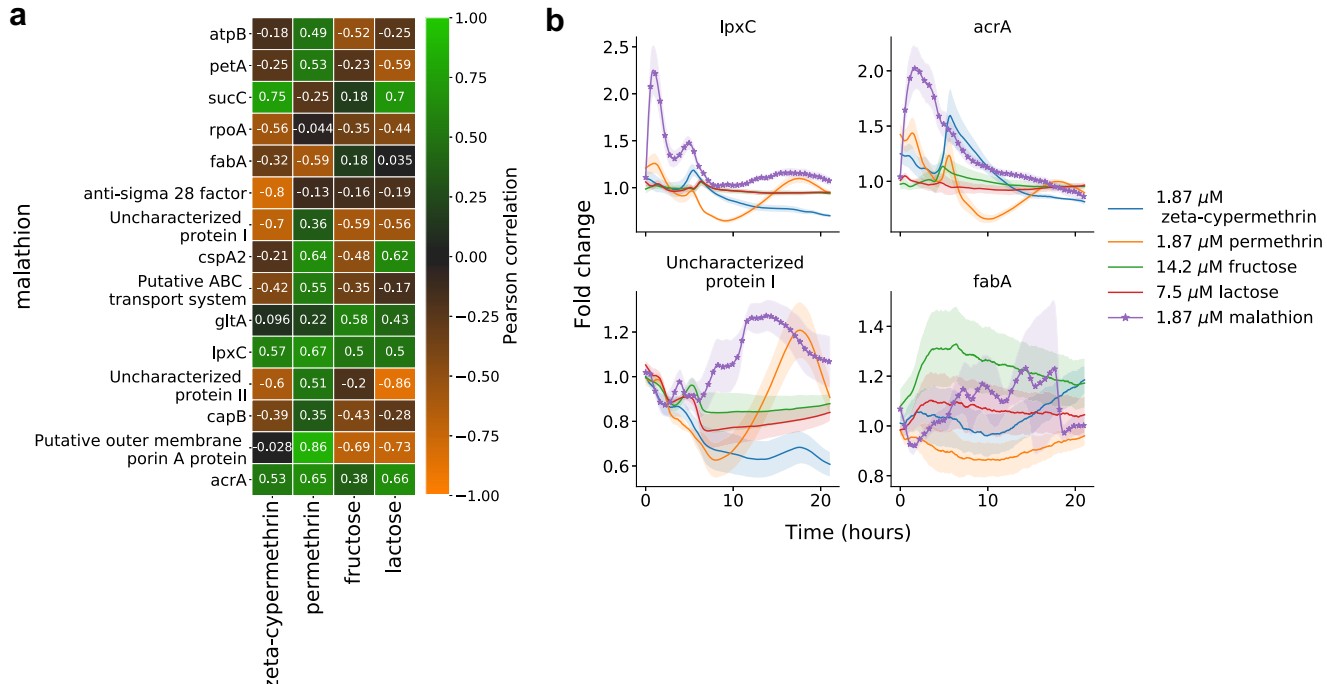

**Fig. 5 | The 15-dimensional genetic reporter cell state provides a unique response to malathion.** a For each genetic reporter, the heatmap depicts the Pearson correlation of the malathion fold change response (rows) with the fold change response to zeta-cypermethrine, permethrin, fructose, or lactose (columns). b The fold change response (reporter + compound with respect to reporter + no compound) of four reporters – two with highest overall correlation and two with lowest overall correlation across compounds. The error bars represent the propagated standard deviations of each of the individual responses across three biological replicates. Source data are provided as a Source Data file.

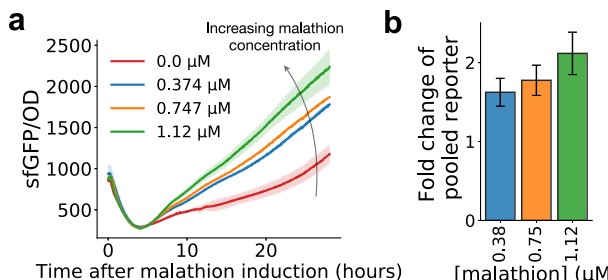

**Fig. 6 | Pooling all 15 malathion reporters results in enhanced reporting for environmental monitoring.** a Time-lapse response after pooling all 15 malathion reporters into a single well and inducing with malathion. Error bars represent the sample standard deviation across three biological replicates. b The fold change at 24 h of the pooled reporter with malathion induction with respect to the pooled reporter without malathion induction. The error bars represent the propagated standard deviations of each of the individual responses across three biological replicates and the bar height represents the fold change calculated using the mean sfGFP/OD signal from malathion and from control conditions. Source data are provided as a Source Data file.

by any individual reporter at the same concentration is 2.3 and is a transient response (*lpxC*, see Fig. 5b). The maximum fold change corresponding with a sustained response is 1.5 obtained by *cspA2*. For sustained response to malathion, the pooled reporter provides more salient response than any individual reporter alone.

Our experiments confirm the usefulness of the malathion reporters outside of the laboratory and in field environments. A potential strategy for environmental malathion monitoring would be to collect a soil sample, culture the pooled reporters from a media made from the sample, then measure the *sfGFP* response. Though this strategy is

enticing, there may be growth competition effects between strains that we have not addressed. Next, we aim to understand if it is possible to detect malathion in environmental samples from our individual reporters.

**Detecting malathion in environmental samples**

The malathion reporter library, selected through observability analysis, has only been examined in an ideal laboratory scenario with either pure or processed malathion whose mass spectrum has been analyzed; it is not yet known if the reporters will be able to sense malathion when induced with actual environmental water samples that have been treated with the insecticide. In the previous section we showed that pooled reporters act as salient malathion sensors. However, confounding factors may be present in the environmental sample such as other small compounds that may make it difficult to deconvolve malathion response from the response due to the confounder. Therefore, in this section we describe an experiment to assess whether or not the malathion concentration can be deduced from our individual reporters treated with environmental insecticide samples.

In order to test if the genetic reporters can sense malathion from environmental samples, irrigation water was collected from three crops after being sprayed with a mixture of Spectracide (50% malathion) and water (Fig. 7a). The concentration of the mixture sprayed was either 0, 1, or 8 times the maximum recommended working concentration of Spectracide – 1 fluid ounce per gallon of water. To rid the solution of unwanted microbes and particles, the irrigation water was strained and filtered prior to to the induction of the genetic reporters (see "Methods" section). The growth and induction protocols all remain the same as for the samples treated with Spectracide in Fig. 4c, d.

We found that a total 9 out of the 15 of the reporters were activated by induction of the irrigation water containing malathion. Figure 7a shows the average per cell fluorescence 24 h after induction of

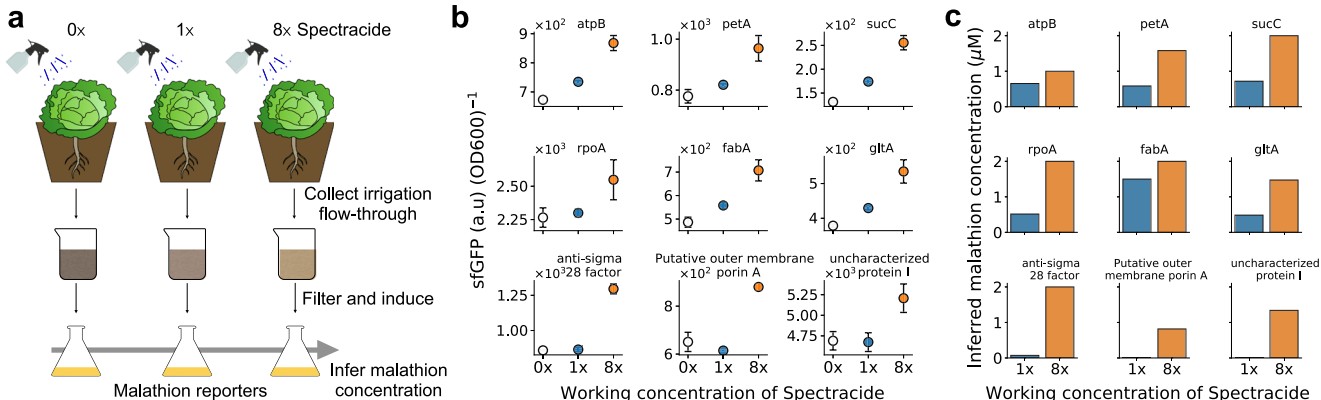

**Fig. 7 | Irrigation water containing malathion from an agricultural setting activates transcriptional reporters and allows for inference of environmental malathion concentration. a** Three cabbage plants are sprayed with a solution of 0, 1, and 8 times the working concentration of Spectracide, respectively. The flow-through is first captured and filtered and then used to induce transcriptional activity in the malathion reporter strains. Using previously characterized response curves for each reporter, an inference for the malathion concentration can be made. **b** The average per cell fluorescence (arbitrary units) of 9 out of the 15 malathon reporters, after 24 h of induction, showed activation due to the soil runoff solution containing malathion. The working concentration of Spectracide is instructed as 1oz of Spectracide to 1 gallon of water. The error bars represent the sample standard deviation from the mean across three biological replicates. Source data are provided as a Source Data file. **c** The concentration of malathion present in the irrigation water is inferred using the signal from **b** and the fitted response curves from Fig. 4d.

the nine strains subjected to 0, 1, or 8 times the working concentration of Spectracide. The reporters *atpB*, *petA*, *sucC*, *rpoA*, *fabA*, and *gltA* all show a response to malathion at 1x working concentration, while the remaining three did not show significant differences from the negative control in this range. Among the strains in Fig. 7b, the strain *sucC* was activated the most, showing an 80% increase from the 0x to 8x condition after the 24 h time period. This shows that many of the selected genetic reporters, 60%, are able to detect malathion in environmentally relevant scenarios, and, furthermore, we can use this data to infer the concentration of malathion present in the samples collected from the environment.

The response curves characterized previously in Fig. 4d for each of the genetic reporters can be used to make an inference about the amount of malathion present in each environmental sample. Note that we are making the assumption that the response curves characterized for each of the nine reporters can be applied to this new setting of treatment with irrigation water. With this assumption we can then use the fitted Hill equations from Fig. 4d and numerically estimate the malathion concentration that reproduces the signal at 1 or 8 times the working concentration of Spectracide. The results obtained are shown in Fig. 7b for each of the nine strains. Through this approach, the reporters provide a range of inferred malathion concentrations; at the working concentration of Spectracide, we can infer that the concentration of malathion is in the range $0.48 - 0.97\,\mu\mathrm{M}$ and at 8 times the working concentration of Spectracide, we can infer the concentration of malathion to be in the range $0.82–2\,\mu\mathrm{M}$. It is important to note that for most, if not all, of the characterized reporter strains, $2\,\mu\mathrm{M}$ was the maximum discernable concentration before the signal saturates. Therefore, it is possible the concentration of malathion is higher than $2\,\mu\mathrm{M}$, however that range cannot be detected by our reporter library.

## Discussion

It is often the case that biologists seek to identify key genes which show variation for the biological process of interest. Many tools have been developed or adapted to meet this need e.g. differential expression, principal component analysis, and gene regulatory network reconstruction to name only a few. However, when using the current tools, there is potential to measure features that are redundant which can lead to wasted time and resources. Furthermore, traditional tools do not provide the capability of optimal gene selection for downstream targeted gene profiling. Therefore, we developed an efficient method that ranks the features for optimal gene selection. The method combines dynamic mode decomposition (DMD) and observability of dynamical systems to provide a systematic approach for the discovery of genes which act as biomarkers for the perturbation-inducible cell state. To extract optimal perturbation sensitive promoters from our model, we showed that genes which contribute highly to observability inform the design of transcriptional reporters that exhibit condition specific sensing.

We introduced DMD as a novel tool for analysis of transcriptome dynamics. In this case, we studied bulk transcriptome dynamics at the minutes resolution and showed that the low-dimensional DMD representation accurately predicts the dynamics and clusters genes based on temporal behavior. Our results suggest that DMD is a capable tool for analysis of transcriptomic data and warrants further exploration in single-cell RNA-seq and other 'omics technologies that aim to infer cell trajectories, pseudotime, and single-cell regulatory networks.

The identification of transcriptional genetic sensors was posed as a design challenge, where a subset of genes are selected to maximize the observability of the cell state. It was shown that a large fraction of genes contribute insignificantly to the cell state observability when only few time points are measured, further validating the common knowledge that genetic networks possess redundancies and are noisy. We also showed that it is significantly more beneficial to measure a sparse set of genes for more time points than to measure more genes for fewer time points. Our results suggest future joint experimental and computational approaches which limit the amount of resources required to get a full description of the system dynamics. A natural extension of our work is to determine how well measurements from a small library of reporters recapitulate the bulk cell state under unseen conditions. Such studies will inform how RNA-seq data should be collected in the future in order to maximize the reconstruction accuracy and minimize labor and experimental costs.

The machine learning driven selection of genetic reporters was shown to produce 15 functional genetic reporters with a variety of malathion dose-response curves. We demonstrated how to aggregate information from each reporter to create a pooled reporter. Moreover, we showed that the genetic reporters can be used to detect malathion in environmental settings, closing the design-build-test loop. More generally, our results and methodology offer an innovative approach that can be used to to identify perturbation-inducible gene expression

systems. We emphasize that our approach takes advantage of the largely untapped resources present in native host genomes and we anticipate that techniques like the one developed here will accelerate the optimization of parts for synthetic biologists to build useful devices from.

Our approach makes no assumptions on the nature of the underlying system. In that sense, the framework we have developed is general and can be applied to data generated from other 'omics techniques and from any organism. In the case that a linear response model is insufficient for capturing the transcriptome dynamics, it can be extended to a variety of nonlinear models to capture nonlinear modes of response[56,66].

Due to only analyzing the transcriptome of SBW25 under specific environmental conditions, our approach cannot guarantee that the identified sensor promoters respond directly to the target analyte of interest. Our approach to biosensing is to view a proxy of the entire cell state, which is a function of the entire underlying network. While this approach is novel, it also implies that the identified sensor promoters may not work in a different host or environmental context. Further refinement of the list of biomarker genes could be obtained by fusing ChIP-seq (chromatin immunoprecipation followed by sequencing) with RNA-seq measurements to discover transcription factors, however such an experimental assay can be prohibitively expensive. The DNA binding sites measured by ChIP-seq alone are not sufficient to infer regulation of transcription. However, together with RNA-seq, the set of biomarkers which causally drive the condition specific response can be uncovered. We envision that our method will accelerate the discovery and design of biosensors in novel host organisms for synthetic biology applications.

## Methods

### Rapid culture sampling
For each biological replicate, *Pseudomonas fluorescens SBW25* glycerol stock was scraped and inoculated in 5 mL of fresh LB broth (Teknova Catalog no. L8022) and was incubated and shaken at 30 °C and 200 r.p.m. for 15 h. The $OD_{600}$ of the 5 mL culture was measured and the entire culture was transferred to 50 mL of fresh LB broth, which was then proceeded by incubation and shaking. Once the $OD_{600}$ of the 50 mL culture reached 0.5, the culture was again passaged into 300 mL of fresh LB broth. The 300 mL culture was grown until $OD_{600}$ of 0.5. Then the culture was split into two 150 mL cultures (one for malathion induction and one for the negative control). The two cultures were sampled at evenly spaced intervals in time (see Supplementary Table 1 for sampling volumes and times) and after the 0 minute sample, malathion (Millipore Sigma Catalog no. 36143) was introduced to the positive condition at 1.83 mM. To separate the media from the cells, a vacuum manifold with 3D printed filter holders was constructed and utilized (Supplementary Fig. 13). In all, 0.45 μm PVDF membrane filters (Durapore Catalog no. HVLP04700) were placed on the filter holders, a vacuum pump was turned on, and the culture sample was dispensed onto the center of the filter, quickly separating the media from the cells. The filter with the cells was then placed into a 50 mL conical centrifuge tube (Fisher Scientific 1495949A) using sterile tweezers. The tube with the filter was then submerged into a liquid nitrogen bath for 10 s to flash freeze the sample. The sample were then stored −80 °C.

### RNA extraction
To extract the RNA, first the filter-harvested cells were resuspended in 2 mL RNAprotect Bacterial Reagent (Qiagen Catalog no. 76506), then pelleted in a centrifuge. To lyse the cells, the pellet was then resuspended in 200 μL of TE Buffer containing 1 mg/mL lysozyme. The RNA was then extracted from the lysed cells using Qiagen RNeasy Mini Kit (Catalog no. 74104), and the samples were DNase treated and concentrated using Zymo RNA Clean and Concentrator (Catalog no. R1019).

### RNA library preparation and sequencing
Bacterial rRNA was depleted using NEBNext Bacterial rRNA Depletion Kit (Catalog no. E7850X). The indexed cDNA library was generated using NEBNext Ultra II Directional RNA Library Prep (Catalog no. E7765L) and NEBNext Multiplex Oligos for Illumina (Catalog no. E6609S). In total, 40 samples (two biological replicates, 10 time points, two conditions) were prepped and sequenced. The library was sequenced at the Genetics Core in the Biological Nanostructures Laboratory at the University of California, Santa Barbara on an Illumina NextSeq with High Output, 150 Cycle, paired end settings.

### Pre-processing of sequencing data
The raw reads were trimmed for adapters and quality using `Trimmomatic`[67]. The reads were then pseudoaligned with `Kallisto`[68] to the *Pseudomonas fluorescens SBW25* transcriptome generated using `GFFRead`[69] and GenBank genome AM181176.4. The normalized gene expression of transcripts per million (TPM), which takes into account sequencing depth and gene length, are used for modeling and analysis. Genes with an average TPM less than 100 in all experimental conditions were discarded from our analysis.

### Malathion reporter library cloning
For the reporter plasmid cassette design, first, the closest intergenic region to the gene target larger than 100 base pairs (bp) was identified based on the open reading frame of the sequenced genome of *Pseudomonas fluorescens* SBW25 (GenBank genome AM181176.4). Primers were designed to include the entire intergenic region in order to capture any transcription-regulator binding sites surrounding the promoter (Fig. 4a). The identified intergenic regions were amplified using the primers and this is what we refer to as 'promoter regions' following the terminology of ref. 70. The promoter regions were cloned into a cassette on the plasmid backbone pBHVK (Supplementary Fig. 8) containing a bicistronic ribosome binding site and super folder GFP (*sfGFP*) as the reporter gene. Lastly, a cloning site was placed in the cassette so that the cloned promoter controls transcriptional activity of *sfGFP*.

The promoters were assembled onto the plasmid backbone pBHVK (see Supplementary Fig. 8) via Golden Gate Assembly[71] using NEB Golden Gate Assembly Kit (Catalog no. E1601S). Because of the potential of arcing during electrotransformation of *Pseudomonas fluorescens SBW25* with Golden Gate reaction buffers, the plasmids are first subcloned into *E. coli Mach1* (Thermo Fisher Scientific Catalog no. C862003) following the manufacturer's protocol for chemical transformation. Between three and six colonies are selected for each strain and the reporter cassette was sent for sequencing at Eurofins Genomics. Then the plasmid DNA was prepared from cultures of transformed Mach1 cells using Qiagen Spin Miniprep Kit (Catalog no. 27106) followed by chemical transformation into *SBW25*. *SBW25* was made chemically competent by washing a culture at $OD_{600}$ of 0.3 with a solution of 10% glycerol two times, then resuspending in 500 μL of 10% glycerol. The plasmid DNA is added to 80 μL of the cell suspension and kept at 4 °C for 30 min, then the cells were electroporated with 1600 V, 200 Ω, and 25 μF. The cells were immediately resuspended in 300 μL of SOC Broth (Fischer Scientific Catalog No. MT46003CR), recovered for 2 h at 30 °C in a shaking incubator, and plated onto 1.5% LB Agar plates with 50 $\frac{\mu g}{mL}$ Kanamycin. Again, three to six colonies of each strain have their reporter cassette sequenced at Eurofins Genomics and simultaneously glycerol stocks of each colony is prepared for long-term storage.

All promoter sequences and plasmid cassette sequences are provided in the Supplementary Data File.

### Photobleaching of spectracide
Spectracide malathion insect spray concentrate (Spectracide Catalog no. 071121309006) was utilized as the environmentally relevant source

of malathion for the reporter library testing and contains 50% malathion. Spectracide is an opaque liquid. We found that we can remove the opaque substances by photobleaching a 5% Spectracide solution (in LB) in a Synergy H1 plate reader (Biotek), at 30 °C and 800 r.p.m. $OD_{600}$ and fluorescence (excitation 485 nm, emission 528 nm) were measured every 3 min for 8 h. To ensure malathion remained in solution after photobleaching, the mass spectrum was analyzed at the University of California, Santa Barbara Mass Spectroscopy Facility. From this we determined that malathion is stable for the course of the photobleaching (Supplementary Figs. 14–26).

## Plate reader assays to measure response curves and doubling times

Scrapes of culture from glycerol stocks of each strain were used to inoculate 3 mL of LB (Kanamycin 50 $\frac{\mu g}{mL}$) in 10 mL 24 deep-well plate sealed with a breathable film (Spectrum Chemical Catalog no. 630-11763) and grown at 30 °C overnight in a shaker incubator. The overnight cultures were diluted to an $OD_{600}$ of 0.1 in 2 mL of LB and the cultures were grown for an additional 2 h. In all, 250 μL of this culture was then transferred to a 96-well optically-transparent microtiter plate. Photobleached spectracide (50% malathion) is then introduced (if relevant) to the cultures in the wells to give the desired concentration of malathion, and grown in a Synergy H1 plate reader (Biotek), at 30 °C and 800 r.p.m. $OD_{600}$ and *sfGFP* (excitation 485nm, emission 528nm) was measured every 3 min for 48 h. Each data point in a response curve was generated by normalizing the *sfGFP* signal (arbitrary fluorescence units) by the $OD_{600}$ to give the average per cell fluorescence, and only the data points before cell death (due to nutrient depletion or media evaporation) are used. The strain growth rates were calculated as $\ln(\text{initial } OD_{600}/\text{final } OD_{600})/(t_{\text{final}} - t_{\text{initial}})$, where the initial $OD_{600}$ is the first measurement within the exponential phase and final $OD_{600}$ is the last measurement within the exponential phase. Then the strain doubling times were calculated as ln(2) divided by the growth rate.

## Collection and cleanup of irrigation water treated with Spectracide

Three cabbage plants were each potted in 5 gallon buckets with fresh soil and a water catchment tray was placed under the plants to catch flow through. The first plant was sprayed with water containing no malathion and the flow through was collected in a 1 L pyrex bottle. The second plant was sprayed with a Spectracide (50% malathion) solution at a concentration of 1 fluid ounce per of gallon water – the maximum working concentration of Spectracide as recommended by the manufacturer. Lastly, the third plant was sprayed with the solution at 8 fluid ounces per gallon of water. Each plant was sprayed for one minute and the collected flow through from each plant were first strained using a 40 μm cell strainer (VWR 76327-098) to remove large microorganisms and large particles. The strained samples were then centrifuged to separate dense, soil particles from the Spectracide solution. Finally, the supernatant was vacuum filtered through a 0.22 μm membrane before induction of the reporters. The protocol for induction of the reporters with the irrigation water is the same as above.

## Computing the dynamic mode decomposition

We now discuss the details of applying dynamic mode decomposition (DMD) to time-series data obtained from sequencing. As mentioned previously, many algorithms have been developed to compute the DMD modes, eigenvalues, and amplitudes, and a key requirement of almost all of the techniques is that the time points are spaced uniformly in time. In our work we begin by collecting the data for a single experimental condition into a time-ordered matrix, **X**, which contains a total of $m \times r$ data snapshots for a data set with $m$ time points and $r$ replicates. For response to malathion, each $\mathbf{x}_i^{(j)}$ corresponds to the

gene expression vector at time $i$ in replicate $j$ and is in the $((i + m) \times j)$th column of the data matrix **X** where $i \in \{0, 1, ..., m - 1\}$ and $j \in \{1, 2, ..., r\}$. For gene expression data obtained from RNA-seq, each data snapshot typically contains thousands of rows denoted by $n$. The $n \times rm$ data matrix for the response to malathion is then given by

$$\mathbf{X}_{\text{malathion}} = \begin{bmatrix} | & | & & | & | & & | \\ \mathbf{x}_0^{(1)} & \mathbf{x}_1^{(1)} & \cdots & \mathbf{x}_{m-1}^{(1)} & \mathbf{x}_0^{(2)} & \cdots & \mathbf{x}_{m-1}^{(2)} & \cdots \\ | & | & & | & | & & | \end{bmatrix} \quad (5)$$

where each $\mathbf{x}_i \in \mathbb{R}^n$ represents the gene expression given in transcripts per million (TPM) from the malathion condition. Similarly, the data matrix for the control condition is constructed. The fold change data matrix, **Z**, is subsequently computed as $\mathbf{Z} = \mathbf{X}_{\text{malathion}} \oslash \mathbf{X}_{\text{control}}$, where $\oslash$ denotes the Hadamard (element-wise) division of two matrices. Next we compute the mean-subtracted and standard deviation-normalized data matrix $\bar{\mathbf{Z}}$

$$\bar{\mathbf{Z}} = \begin{bmatrix} \frac{\mathbf{z}_0 - \boldsymbol{\mu}_{0:m-1}}{\boldsymbol{\sigma}_{0:m-1}^2} & \frac{\mathbf{z}_1 - \boldsymbol{\mu}_{0:m-1}}{\boldsymbol{\sigma}_{0:m-1}^2} & \cdots & \frac{\mathbf{z}_{m-1} - \boldsymbol{\mu}_{0:m-1}}{\boldsymbol{\sigma}_{0:m-1}^2} \end{bmatrix} \quad (6)$$

where $\boldsymbol{\mu}_{0:m-1}$ is the vector of time-averages of each gene and $\boldsymbol{\sigma}_{0:m-1}^2$ is the vector of time-standard deviations of each gene. The divisions in Eq. (6) are performed element-wise. We see that $\bar{\mathbf{Z}}$ is obtained by removing the time-averages from each gene and standardizing the time-variances of each gene. The mean subtraction operation is motivated by the fact that the mean of the data corresponds to the eigenvalue $\lambda = 1$, which is always an eigenvalue of the Koopman operator, the operator that DMD ultimately aims to approximate[72], and not one we are particularly interested in. The normalization by the standard deviation is performed so that the magnitude of the fold change has no implication on the connectivity of the learned dynamical system.

The algorithm we make use of to compute the dynamic mode decomposition (and the approximation of the Koopman operator) is exact DMD[37], which aims to identify the best-fit linear relationship between the following time-shifted data matrices

$$\bar{\mathbf{Z}}_p = \begin{bmatrix} \bar{\mathbf{z}}_0 & \bar{\mathbf{z}}_1 & \cdots & \bar{\mathbf{z}}_{m-2} \end{bmatrix}, \qquad \bar{\mathbf{Z}}_f = \begin{bmatrix} \bar{\mathbf{z}}_1 & \bar{\mathbf{z}}_2 & \cdots & \bar{\mathbf{z}}_{m-1} \end{bmatrix}$$

such that

$$\bar{\mathbf{Z}}_f = \mathbf{K}\bar{\mathbf{Z}}_p + \mathbf{r} \quad (7)$$

where **r** is the residual due to **K** only providing an approximation of the actual dynamics. Note that there are $n^2$ unknown parameters in **K** and $n \times m$ equations in Eq. (7). The residual is then minimized by Exact DMD (in the least squares sense) by first considering the reduced singular value decomposition (SVD) of $\bar{\mathbf{Z}}_p = \mathbf{U}\boldsymbol{\Sigma}\mathbf{T}^\top$ where $\boldsymbol{\Sigma} \in \mathbb{R}^{k \times k}$. As the number of time points, $m$, obtained from sequencing is typically much less than the number of genes, $n$, we keep $k \leq m$ singular values. Recognizing that minimizing the residual requires it to be orthogonal to the left singular vectors, we can pre-multiply (7) with $\mathbf{U}^\top$ to obtain

$$\mathbf{U}^\top \bar{\mathbf{Z}}_f = \mathbf{K}\mathbf{U}\boldsymbol{\Sigma}\mathbf{T}^\top. \quad (8)$$

Rearranging the above equation, it is shown that **K** is related to $\hat{\mathbf{K}}$ through a similarity transformation as shown in Eq. (9)

$$\hat{\mathbf{K}} = \mathbf{U}^\top \bar{\mathbf{Z}}_f \mathbf{T}\boldsymbol{\Sigma}^{-1} = \mathbf{U}^\top \mathbf{K}\mathbf{U} \quad (9)$$

meaning that the eigenvalues of $\hat{\mathbf{K}}$, $\lambda$, are equivalent to the $k$ leading eigenvalues of **K** while the eigenvectors of $\hat{\mathbf{K}}$, **s**, are related to the $k$ leading eigenvectors of **K**, **v**, by $\mathbf{v} = \mathbf{U}\mathbf{s}$. This eigendecomposition then allows the fold change response to be written as the following spectral

decomposition

$$\hat{\mathbf{z}}_i = \sum_{j=1}^{k} \mathbf{v}_j \lambda_j^i \mathbf{b}_j = \mathbf{V} \Lambda^i \mathbf{b} \tag{10}$$

where $\mathbf{V}$ is a matrix whose columns are the eigenvectors (DMD modes) $\mathbf{v}_j$, and $\mathbf{b}$ is a vector of amplitudes corresponding to the gene expression at the initial time point as $\mathbf{b} = \mathbf{V}^\dagger \hat{\mathbf{z}}_0$. Here $\dagger$ represents the Moore-Penrose pseudoinverse of a matrix.

Using the above spectral decomposition, the modes can then be evolved in time for $m-1$ time steps to reconstruct the data from knowledge of the initial condition. Evolving past the $m$th time point allows for forecasting of the fold change response. To measure the accuracy of reconstruction we use the coefficient of determination

$$R^2 = 1 - \frac{\sum_{i=0}^{m}(\hat{\mathbf{z}}_i - \tilde{\mathbf{z}}_i)}{\sum_{i=0}^{m}(\hat{\mathbf{z}}_i - \bar{\mathbf{z}})} \tag{11}$$

where $\bar{\mathbf{z}}$ is the vector of each gene's mean expression, formally $\bar{z}^{(j)} = \sum_{k=0}^{m} \hat{z}_k^{(j)}$, and $\tilde{\mathbf{z}}_k = \mathbf{K}^k \hat{\mathbf{z}}_0$ is the prediction of $\hat{\mathbf{z}}_k$ given by the model starting from the initial condition.

## Computing the gene sampling weights

Here we describe our methodology for ranking genes based on their contribution to the observability of the dynamical system learned via dynamic mode decomposition. We start by introducing the energy of a signal in discrete-time as

$$E_y = \sum_{i=0}^{\infty} \mathbf{y}_i^\top \mathbf{y}_i \tag{12}$$

which is closely related to the idea of energy in the physical sense and where $\mathbf{y} = \mathbf{W}\bar{\mathbf{z}}$ are measurements of the system state and $\mathbf{W} \in \mathbb{R}^{p \times n}$. Rewriting the signal energy (12) using the recursion for $\mathbf{y}$ given as $\mathbf{y}_t = \mathbf{W}\mathbf{K}^t\bar{\mathbf{z}}_0$, we can reveal the connection between energy and observability

$$\begin{aligned} E_y &= \sum_{i=0}^{\infty} \bar{\mathbf{z}}_0^\top \mathbf{K}^{i^\top} \mathbf{W}^\top \mathbf{W} \mathbf{K}^i \bar{\mathbf{z}}_0 \\ &= \bar{\mathbf{z}}_0^\top \left( \sum_{i=0}^{\infty} \mathbf{K}^{i^\top} \mathbf{W}^\top \mathbf{W} \mathbf{K}^i \right) \bar{\mathbf{z}}_0 \\ &= \bar{\mathbf{z}}_0^\top \mathcal{X}_o \bar{\mathbf{z}}_0 \end{aligned} \tag{13}$$

where $\mathcal{X}_o$ is the infinite-horizon observability gramian, a symmetric matrix that is unique if the eigenvalues of $\mathbf{K}$ all have magnitude <1. The observability gramian describes how much gain will be attained by a system's output, $\mathbf{y}$, given an initial condition $\bar{\mathbf{z}}_0$. It simultaneously gives a measure of how well the initial condition $\bar{\mathbf{z}}_0$ can be estimated given only measurements of the system state $y$[61].

We use the observability gramian along with the measure of energy it provides to optimize for the gene sampling weights in the rows of $\mathbf{W}$ that maximize the signal energy $E_y$. Formally, the objective function is given as

$$\begin{aligned} &\max_{\mathbf{W} \in \mathbb{R}^{p \times n}} \bar{\mathbf{z}}_0^\top \mathcal{X}_o \bar{\mathbf{z}}_0 \\ &\text{subject to } \mathbf{W}\mathbf{W}^\top = I_{p \times p}. \end{aligned} \tag{14}$$

where we seek the matrix $\mathbf{W}$ that maximizes the observability of the cell state $\bar{\mathbf{z}}_0$. The constraint above enforces the following three points, (i) the length of each row vector in $\mathbf{W}$ is not important, we are only

concerned with the direction and the constraint sets the length of each row vector to be equal to 1, (ii) the maximization problem is well-posed, i.e. the objective cannot blow up to infinity with the length constraint, and (iii) the rows of $\mathbf{W}$ form $p$ vectors of an orthonormal basis for $\mathbb{R}^p$, i.e. $\mathbf{W}\mathbf{W}^\top = I_{p \times p}$. Each row vector in $\mathbf{W}$ can then be viewed as a set of weights, each orthogonal to one another, that rank genes based on their contribution to the observability of the system. The optimization problem (14) represents a quadratic program with linear constraints, and the rows of $\mathbf{W}$ which maximize the objective are the $p$ eigenvectors corresponding to the $p$ eigenvalues with highest magnitude of the Gram matrix

$$\mathbf{G} = \sum_{i=0}^{\infty} \mathbf{K}^i \bar{\mathbf{z}}_0 \bar{\mathbf{z}}_0^\top \mathbf{K}^{i^\top}. \tag{15}$$

Since $\mathbf{G} \in \mathbb{R}^{n \times n}$ is a sum of quadratic forms, the result is that $\mathbf{G}$ has non-negative, real-valued eigenvalues. If the eigendecomposition is $\mathbf{G} = \mathbf{Q}\mathbf{D}\mathbf{Q}^{-1}$, then the solution to the optimization problem Eq. (14) is

$$\mathbf{W} = \begin{bmatrix} \mathbf{q}_1^\top \\ \vdots \\ \mathbf{q}_p^\top \end{bmatrix} \tag{16}$$

where $\mathbf{q}_1$ through $\mathbf{q}_p$ are the top eigenvectors of the Gram matrix $\mathbf{G}$. The proof of the solution to the optimization problem is provided in the Supplementary Information (Section 1.1). The single set of gene sampling weights that maximize the observability are precisely $\mathbf{q}_1$ and from here on out we call these weights $\mathbf{w}$.

Since transcriptomic data sets typically have few initial conditions, i.e. biological and technical replicates, before solving for $\mathbf{w}$ we enrich our data set with $N$ synthetic initial conditions that are randomly sampled as $Uniform(\min(\bar{\mathbf{z}}_0^{(j)}), \max(\bar{\mathbf{z}}_0^{(j)}))$ where $j$ in $\{1, 2, \ldots, r\}$ and $r$ is the number of replicates. The motivation for the artificial data generation is given in ref. 73, where it is shown that artificially generated data points improved the estimate of the DMD model when the data set is affected by noise. $N$ is chosen to be equal to the number of genes to ensure the matrix of initial conditions has full rank. Another issue that we have addressed are the instabilities present in the DMD eigenvalues. Consequently, the observability gramian is not unique and the sum in Eq. (15) diverges to infinity. To mend this issue, we compute the finite-horizon Gram matrix, where the sum in Eq. (13) and Eq. (15) is from 0 to $m$. This allows for the computation of the finite-horizon signal energy from Eq. (13) where the bounds on the sum are now from $i = 0$ to $i = m$.

Once $\mathbf{w}$ is obtained by solving Eq. (14), then measurements $y_t$, for $t$ in $\{0, 1, \ldots, T\}$, are generated from $y_t = \mathbf{w}^\top \mathbf{K}^t \bar{\mathbf{z}}_0$ while keeping only the $q$ elements of $\mathbf{w}$ with largest magnitude as nonzero. All other elements of $\mathbf{w}$ are set to zero to simulate the sampling of only selected genes. To reconstruct $\bar{\mathbf{z}}_0$ using only the measurements, we form the following observability matrix from the known sampling weights, $\mathbf{w}$ and the dynamics matrix $\mathbf{K}$

$$\begin{bmatrix} y_0 \\ y_1 \\ y_2 \\ \vdots \\ y_T \end{bmatrix} = \begin{bmatrix} \mathbf{w}^\top \\ \mathbf{w}^\top \mathbf{K} \\ \mathbf{w}^\top \mathbf{K}^2 \\ \vdots \\ \mathbf{w}^\top \mathbf{K}^T \end{bmatrix} \bar{\mathbf{z}}_0 = \mathcal{O}_T \bar{\mathbf{z}}_0 \tag{17}$$

and using the Moore-Penrose pseudoinverse we can obtain an estimate of the initial condition as follows

$$\mathcal{O}_T^\dagger \begin{bmatrix} y_0 \\ y_1 \\ y_2 \\ \vdots \\ y_T \end{bmatrix} = \hat{\mathbf{z}}_0 \approx \bar{\mathbf{z}}_0. \quad (18)$$

Increasing $q$ while keeping $T$ constant results in increasing reconstruction accuracy until a critical value of $q$ such that the reconstruction accuracy plateaus; a similar scenario holds for keeping $q$ constant and increasing $T$. When both $T$ and $q$ surpass the critical values, perfect reconstruction may be achieved.

When the computation of the Gram matrix, $\mathbf{G}$, is not computationally feasible, as can be the case when the dimensionality of the data are relatively high compared to that of bacterial transcription networks that we are dealing with here, the reduced order dynamics given by DMD can be used to compute an approximation to the leading eigenvalues and eigenvectors. The reduced order $\mathbf{G}$ is then given by

$$\tilde{\mathbf{G}} = \sum_{i=0}^{\infty} \hat{\mathbf{K}}^i \mathbf{U}^\top \bar{\mathbf{z}}_0 \bar{\mathbf{z}}_0^\top \mathbf{U} \hat{\mathbf{K}}^{i\top} \quad (19)$$

where $\hat{\mathbf{K}}$ and $\mathbf{U}$ are given in Eq. (9). Supplementary Fig. 4 shows the approximation of the leading eigenvalues and eigenvectors of $\mathbf{G}$ by $\tilde{\mathbf{G}}$.

### Reporting summary
Further information on research design is available in the Nature Portfolio Reporting Summary linked to this article.

## Data availability
The RNA sequencing data generated in this study have been deposited in the GEO database under accession code GSE200822. The processed RNA sequencing data, DNA sequencing, and plate reader data are available at the Github repository https://github.com/AqibHasnain/transcriptome-dynamics-dmd-observability in the data directory. The plate reader data used in this study are also provided in the Source Data file. Sequences of oligonucleotides are provided in the Supplementary Data file. Source Data are provided as a Source Data file. Source data are provided with this paper.

## Code availability
All codes used in this study are available at: https://github.com/AqibHasnain/transcriptome-dynamics-dmd-observability or available from the author's upon request. The git repo hash key associated with this manuscript is a0b742e.

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

## Acknowledgements

This work was supported in part by a contract awarded by the Defense Advanced Research Projects Agency and the Air Force Research Laboratory under the research program "Synergistic Discovery and

Design (SD2)", through grant numbers FA8750-17-C-0229, HR001117C0092, HR001117C0094, and DEAC0576RL01830. This work was supported in part by a subcontract awarded by the Pacific Northwest National Laboratory for the Secure Biosystems Design Science Focus Area "Persistence Control of Engineered Functions in Complex Soil Microbiomes" sponsored by the U.S. Department of Energy Office of Biological and Environmental Research and Pacific Northwest National Laboratories, grant number 545157. This work was also supported in part by an Army Research Office Young Investigators Program Award with grant number W911NF-20-1-0165 and Army Research Office grants administered through the Institute of Collaborative Biotechnologies, with grant numbers W911NF-19-0026, W911NF-19-F-0037, and W911NF-19-D-0001. We acknowledge the use of the Biological Nanostructures Laboratory within the California NanoSystems Institute, supported by the University of California, Santa Barbara and the University of California, Office of the President. We thank Ryan Chambers, Trevor Marks, and Kirk Fields for construction of the vacuum manifold. We thank Jamiree Harrison for engaging in insightful discussions on linear systems theory.

## Author contributions

A.H. and E.Y. conceived of the project. A.H. designed research and experiments, developed computational and analytical analysis methodologies, performed formal analysis, analyzed data, and wrote the manuscript. S.B. assisted with RNA-seq sample collection and observability analysis. D.M.J. assisted with cloning of reporter strains; J.S. performed the mRNA library prep and sequencing; S.B.H. assisted in conceptualization and designing the time-series RNA-seq experiment; E.Y. supervised research and secured funding. A.H. revised the manuscript with inputs from all authors.

## Competing interests

The authors declare no competing interests.
