## [Peer Review File · Nature Communications]

Reviewers' Comments:

Reviewer #1:

Remarks to the Author:

The authors describe a new mathematical approach for analyzing time series gene expression data that they seek to apply to the discovery of novel biosensors. The method utilizes DMD (well established) to estimate a first order difference equation model from time series gene expression data. The authors then apply a novel method to assign weights to genes to maximize signal energy and measure the contribution of each gene's expression to the observability of the DMD estimated model. They authors also describe how to estimate the initial expression state of the system based on the estimated weights, the DMD model, and expression measurements from subsequent time points. Their results suggest that this approach can build an accurate predictive model – capturing 92% of the variance of the training data. While this mathematical approach is innovative, and could have applications for gene expression analysis, the authors seek to apply the approach to biosensor discovery, and it is in this application that I have major concerns.

Major Concerns

- The approach does not identify biosensors since the elements identified do not sense the biological analyte. Rather, the approach identifies a set of promoters whose expression is (broadly speaking) perturbed by the analyte.
- The promoters identified cannot, by the very definition of a promoter, sense the analyte directly (rare exceptions would include attenuation, but there is no evidence of this in the results)
- Rather, the promoters are responding to changes in other regulatory elements that are somehow being perturbed by the presence of the analyte
- The expression of the promoters thus depends inextricably on the rest of the regulatory network of the host organism. The promoters therefore do not contribute to a "pool of parts" that can be easily used in different contexts for synthetic biology – a primary stated motivation of the project
- Similarly, the weights, W , learned through the observability and signal energy approach are functions of the entire system. It is therefore not clear what relevance they would have or how they would have to be changed when the encoder promoters are used in a different context.
- More broadly, there is nothing in the method as described that can confirm if the profiled organism contains an element that specifically detects the target analyte. More likely, the changes in expression are a host of indirect changes in response to changes in metabolism induced by something about the target analyte.
- This is evident in the genes they identify in Figure 3d, none of which seems to have a direct role in malathion detection or metabolism, but most of which have very broad roles in metabolism
- Specificity is thus a very serious concern. Because the promoters identified are only indirectly responding, and most likely due to broad changes in metabolism, many metabolites that result in similar metabolic alterations are likely to perturb the encoder promoters. There is virtually no testing of the specificity of the response.
- Even sensitivity is an issue, for all of the above reasons. Because the encoder promoters are responding to many broad metabolic signals, changes in external conditions will impact their response and utility for detecting any single target analytes.
- This is clearly reflected in the results of testing they performed on environmental samples. Only 9 of 15 handpicked reporter promoters were activated by the target analyte in a different context. This is only a little over half of the promoters. And if one assumes that a promoter can either be induced or not, this is not much above chance.

Minor comments

- The authors use the phrase "encoder genes", but it is really the promoters of the genes that are responding. This is important as the genes themselves are not the biosensor elements. (Although a more traditional understanding of the definition of a biosensor would imply a protein that directly interacts with the analyte).
- If I understand correctly, the paper describes creating a "virtual sensor" by exposing each reporter promoter to the sample, and then mathematically aggregating the response. In other words, pooling is done at the data analysis level. How, in practice, could pooling be done at the assay level? This would be far more convenient.
- To select encoder promoters, subsets of 15 random genes are selected from the top half of

genes, as ranked by W . And then the best subset selected. Why this random approach. Why not simply select the top N genes ranked by W ?

- In the methods, it is initially confusing that W is in both the SVD decomposition and the matrix of estimated gene weights. It would be helpful to note somewhere near line 1115 that the W in this section is not the same as the W in the previous one.

- It is not clear how the mathematical approach described is a practical advance for biosensor prediction compared to previous methods more directly designed to target mode of action: T. S. Gardner et al. (2003) *Science*, H. Xing et al. (2006) *Nature protocols*, J. J. Faith et al. (2007) *PLoS biology*

- There have been recently published manuscripts that describing successes in mining for novel biosensors that should be cited: J. Ibero et al. (2021) *Genes (Basel)*, C. Grazon et al. (2020) *Nature communications*

Reviewer #2:

Remarks to the Author:

This paper present a validated method to identify metabolite-responsive biosensors from rna-seq data. It relies on a novel combination of ideas from control theory applied to temporal rna-seq data; the conclusions of the analysis were validated by building 15 biosensors for malathion, a pesticide common employed in crop management.

I read the paper with great interest as I have domain knowledge of control theory and machine learning, as well as experience in temporal RNA-seq data clustering and biosensor design. This is a highly multidisciplinary paper that combines ideas from various fields to address an important challenge. I particularly liked the experimental validation because the analyte in question is a compound of relevance and the reporters were implemented in a non-model host that is an actual soil dweller (as opposed to validations on model organisms that may respond very differently from those that live in the field).

I found the flow of the text and sections a bit cumbersome, and as I result I found myself jumping back and forth quite a bit. So I will first summarise my understanding of the paper. There are five results:

- 1) Temporal RNA-seq data in response to malathion (Figure 1).
- 2) A method to extract a low-dimensional model that can approximate the temporal response of the whole transcriptome (Figure 2).
- 3) A method to rank genes according to their impact on the observability of the low-dimensional model in 2) (Figure 3)
- 4) Experimental validation of 15 genes extracted from 3), which were tagged with a fluorescent reporter to demonstrate the utility of these genes as sensors (Figure 4)
- 5) A sensor fusion technique to blend imperfect readouts from several sensors into a single informative sensor - with experimental validation (Figure 5-6).

The methods combine dynamic mode decomposition (DMD, approximating the Koopman operator) with observability analysis, and as far as I can tell this combination is a novel approach to analyze RNA-seq data and the calculations appear technically correct.

I have two main concerns:

- 1) It is unclear what benefits this approach has over other methods to perform the same analysis. There is little mention or comparisons to other methods to cluster temporal RNA-seq data (e.g. hierarchical clustering based on a similarity score, graph-based clustering or community detection methods, use of bayes factors as gene-to-gene similarity scores; there are several others). Also, a similar analysis could be done with traditional differential expression analysis - I agree with

authors that differential expression analysis was not conceived for temporal data, but most of the genes that their method selects (Fig 3D) are ultimately those that are permanently up or down regulated after the stimulus, and I suspect these would have been picked up with differential expression analysis at $T=8$.

This question is particularly relevant because the paper does use some of these same standard tools later in the paper: Fig 4b contains a hierarchical clustering of 15 genes and it is unclear why that cannot be extended to the original 624 genes. To me, it seems that a unique feature of their Gramian-based approach is the ability to rank genes according to importance (Figure 3A), but as I point out below, there are questions about the effectiveness of this approach.

To be clear, I don't think authors need to benchmark their approach to many other methods from the literature - I suggest instead they clearly highlight the benefits of their method. None of existing methods is perfect, so it would be valuable to clearly state the gaps covered by their method. Their approach is theoretically elegant and nicely combines tools from different disciplines but I suspect the broad readership of the journal will not necessarily appreciate the mathematical novelty and time series clustering is a very well covered subject.

2) I have an observation regarding the methodology itself. Fig 3A shows all 624 genes ranked by weights obtained from the Gramian approach. However, this curve looks almost linear, which unless I am missing something, casts doubts on the selectivity of the method. There is a comment on this in L435-437, but for the approach to be useful, one would expect an S-shaped curve so that a large number of genes have poor scores and few have a high score, and then genes can be selected by imposing a cut-off on the high scoring genes. Such an almost linear relation suggests that the method is poorly selective and cannot discriminate between relevant and not relevant genes. I believe the panel on the right verifies this: for $T=8$ the three groups of genes have very similar reconstruction accuracies; the text suggests that the high scorers (orange) are better but given the overlap between the violin plots, that claim should be backed up by a statistical significance test, e.g. by testing against a null hypothesis with randomly selected genes across all three groups. For $T=2$, there are interesting and clear differences across groups, and maybe this is the regime the authors should focus on, as it seems that the method does pick up differences in reconstruction power across the three groups of genes.

Specific comments:

- I am not sure if the term 'encoder genes' speaks to the audience. These would typically be called biomarkers, responsive genes, or similar.
- L197-L200. Please explain how the 624 genes were selected; the normal approach is to use a volcano plot or similar to select differentially expressed genes (eg with DESeq2 or Limma) above a cutoff p-value; this is standard in rna-seq analysis.
- L204-207: Please explain how this assumption holds when the host metabolizes the analyte, i.e. its intracellular concentration cannot be assumed constant but decays in time.
- L296-297 repeated in L279-280.
- L377-378. Please clarify what 'representative' stands for - does it mean there are only five clusters across the 624 genes?
- L544-551: This is a key point but I could not understand how the 15 genes were selected and why there was a need for a randomization step. Why not select the top 15 genes ranked according to the computed sampling weights? This was done in L532-538 but it is unclear why a different procedure is needed here.
- L495-502: this is an insightful comment but unclear where this can be seen in Fig 3B.
- L607-608: please clarify what is meant by sfGFP being a reporter for transcription initiation.
- L657: Michaelis-Menten kinetics are hyperbolic and do not resemble Hill equations (unless $n=1$).

- Figure 2A is unclear and think 2D should be the main content - I appreciate the effort to make the mathematics more accessible but I don't think end users will understand 2A. The Results does mention mathematical notation, but these are only defined in the Methods, which makes it hard to follow. I suggest to include few equations into the results section to make the text clear and unambiguous.
- Suggest to merge Fig 2B-C because they share the data markers.
- In Figure 3A genes with highest rank have a score of 1, but a score of 0 in 3D.
- Figure 4B: were the gene-to-gene correlations computed at a fixed time, or averaged across all timepoints?
- Figure 4D: From the caption and Table 1 it seems that the dose-response curves were computed at different times, which were determined according to the maximal fluorescence. This is unusual and makes the dose-response curves not comparable to each other. Normally dose-response curves are computed at mid-exponential growth or similar. Moreover, without seeing the growth curves or kinetic data (sfGFP fluorescence/OD) it is impossible to establish if those selected times are representative or artefacts. It is also unclear if the maximal fold change mentioned in L667 is the ratio between sfGFP_sensor/sfGFP_control or $(\text{sfGFP_sensor}/\text{OD_sensor})/(\text{sfGFP_control}/\text{OD_control})$. These issues can have a huge impact on the shape of the dose-response curve, see eg <https://pubs.acs.org/doi/abs/10.1021/acssynbio.2c00143>
- Figure 4D: it would be nice to see the dynamic range and K_m of all sensors in a single scatter plot; if done, it would be important to report error bars for uncertainty estimates given by the fitting procedure; particularly for the Hill coefficients as their fitted values are known to be very sensitive to noise.
- Figures 5-6. I found these results slightly disconnected from the previous and at times read as if they could be the subject of a different paper. I suggest to improve the linkage between the two parts of the paper.
- The computer code in github is well presented, but I'd encourage authors to either lock the repo, or provide the hash key of the commit that corresponds to the paper submission. Otherwise it is impossible to know which version of the code corresponds exactly to what was done in the manuscript. This is not their fault and is part of a wider trend in the field, where code repos are left open and, as a result, it is not possible to establish reproducibility. Another alternative is to simply attach code as a supplementary zip file.

Response to reviewer comments

Aqib Hasnain et al.

Dear Dr. Cloney,

Thank you for giving us the opportunity to submit a revised draft of our manuscript to Nature Communications. We appreciate the time and effort that you and the reviewers dedicated to providing feedback on our manuscript and are grateful for the insightful comments on and valuable improvements to our paper. We have addressed many of the concerns and incorporated most of the suggestions made by the reviewers. Those changes are highlighted within the manuscript. Please see below, in blue, for a point-by-point response to the reviewers' comments and concerns.

Response to Reviewer 1

- *The approach does not identify biosensors since the elements identified do not sense the biological analyte. Rather, the approach identifies a set of promoters whose expression is (broadly speaking) perturbed by the analyte.*
- *The promoters identified cannot, by the very definition of a promoter, sense the analyte directly (rare exceptions would include attenuation, but there is no evidence of this in the results)*
- *Rather, the promoters are responding to changes in other regulatory elements that are somehow being perturbed by the presence of the analyte*

We thank the reviewer for correctly pointing out that we have not identified the causal mechanism in which the perturbation interacts with the genetic, protein, or metabolite network of the cell. Since we only measure transcriptional changes over time, our method cannot identify protein targets of a perturbation. We were imprecise in our use of the term biosensor to describe sensor promoters. We have modified the manuscript to replace “biosensor” with “analyte-responsive promoter” or “genetic reporter”.

We have emphasized our work's main objective to identify a set of responsive, endogenous promoters which can be used as reporters of a perturbation-specific cell state. The underlying objective is to use the reporters to act as a proxy for the entire transcriptome. In the absence of a single transcription factor - promoter pair that directly signals malathion concentration, we envision that a small set of promoters which respond specifically and reproducibly can act as a cell state sensor and therefore as a proxy malathion sensor.

Similarly, we have demonstrated that our analyte responsive promoters change state in response to the analyte. We have shown that 9 of 15 promoters have a highly functional response in a real-world setting. Even though they show a response, we must still test their specificity. If we look at solely steady-state or transient response (single or few time points), each of the promoters may not be specific to malathion. However when looking at the temporal response, we see that malathion induces a unique response among the 15 reporters (see Fig. 5 in revised manuscript and Supplementary Figure 8).

- *The expression of the promoters thus depends inextricably on the rest of the regulatory network of the host organism. The promoters therefore do not contribute to a “pool of parts” that can be easily used in different contexts for synthetic biology – a primary stated motivation of the project*
- *Similarly, the weights, W , learned through the observability and signal energy approach are functions of the entire system. It is therefore not clear what relevance they would have or how they would have to be changed when the encoder promoters are used in a different context.*

- More broadly, there is nothing in the method as described that can confirm is the profiled organism contains an element that specifically detects the target analyte. More likely, the changes in expression are a host of indirect changes in response to changes in metabolism induced by something about the target analyte.

We thank the reviewer for their feedback on context in which our identified parts will function. This limitation of our approach has now been clearly outlined and the scope of our work has been clearly stated in the introduction. We again thank the reviewer for helping us improve on the clarity of the messaging in the manuscript. Specifically, we state that our approach provides no guarantee on the functionality of the sensor promoters outside the context of the host organism *P. fluorescens* SBW25. We highlight that the purpose of our work is to identify subsets of genes which recapitulate the entire cell state, that these genes are endogenous to the host organism, and that due to interdependencies present in reconstructed gene regulatory networks, the genes identified with our developed approach are not guaranteed to act as biosensors for malathion in different contexts (see L104-L120 and L929-L936 in revised manuscript).

- This is evident in the genes they identify in Figure 3d, none of which seems to have a direct role in malathion detection or metabolism, but most of which have very broad roles in metabolism

- Specificity is thus a very serious concern. Because the promoters identified are only indirectly responding, and most likely due to broad changes in metabolism, many metabolites that result in similar metabolic alterations are likely to perturb the encoder promoters. There is virtually no testing of the specificity of the response.

- Even sensitivity is an issue, for all of the above reasons. Because the encoder promoters are responding to many broad metabolic signals, changes in external conditions will impact their response and utility for detecting any single target analytes.

We thank the reviewer for valuable feedback on specificity and sensitivity of the reporters. We did not adequately convey the advantage and motivation of our developed method in the manuscript's first draft. Our method aims to identify the subset of genes which best recapitulate the perturbation-specific cell state from time-series RNA-seq; these datasets inherently necessitate a small amount of biological replicates due to the present cost of sequencing. The hypothesis we propose is that the identified subset of genes should be unique to each distinct perturbation. The genetic reporters we developed can then be seen as a 15-dimensional fingerprint of the malathion perturbation and under the proposed hypothesis, non-malathion perturbations should induce a distinct response in **at least one** of the reporters. This is the weakest result we can hope for and we expect, given the high-dimensionality of regulatory networks, that many of the reporters will produce uncorrelated outputs for new perturbations.

To show this, in our revised manuscript we performed specificity experiments by subjecting the reporters to four other perturbations. One monosaccharide (fructose), one polysaccharide (lactose), and two pyrethroid insecticides were tested and the results are shown in Figure 5 of the revised submission manuscript along with the full time-lapse responses in the supplement. Overall, we found that the reporter responses to the four other perturbations were uncorrelated with the reporter response to malathion. The specificity experiments indicate that the reporters are well suited for malathion detection in the media conditions that we have used.

Furthermore, we agree that sensitivity is an issue with the constructed reporters. Since the response of SBW25 to malathion was only measured in LB broth, a rich nutrient media, it is not expected that the reporters function in, for example, a non-rich nutrient media and we do not claim this to be the case.

- This is clearly reflected in the results of testing they performed on environmental samples. Only 9 of 15 handpicked reporter promoters were activated by the target analyte in a different context. This is only a little over half of the promoters. And if one assumes that a promoter can either be induced or not, this is not much above chance.

We completely agree with the reviewer in their line of thinking that a promoter can either respond or not. There are approximately $N = 3000$ promoters in the SBW25 genome assuming each operon has a single promoter. We can denote the probability of a differential response of promoter P as p . We empirically

characterized p to be 180/3000 or 6% through differential expression analysis ($K = 180$ differentially expressed genes called by DESeq2). Thus, if we were to leave it to chance, we can use a hypergeometric test to model how many promoters, k , of $n = 15$ randomly selected out of $N = 3000$ promoters would exhibit a response. The hypergeometric distribution defines the probability of k successes drawn from n samples (without replacement) from a population N with successes K and is given as

$$p_P(k) = \frac{\binom{K}{k} \binom{N-K}{n-k}}{\binom{N}{n}}.$$

Under the hypergeometric distribution with $k = 9$ successes out of $n = 15$ at the sample level and $K = 180$ successes out of $N = 3000$ at the population level, we arrive at $p_P(9) = 0.9$ promoters out of 15 to be responsive to malathion. Our results are then over-enriched 10-fold compared to expectations with a p-value of $3e - 8$, emphasizing that our results are extreme when comparing to chance.

- It is not clear how the mathematical approach described is a practical advance for biosensor prediction compared to previous methods more directly designed to target mode of action: T. S. Gardner et al. (2003) Science, H. Xing et al. (2006) Nature protocols, J. J. Faith et al. (2007) PLoS biology

We deeply appreciate the reviewer identifying these three critical references from the field of network reconstruction. We carefully studied these papers and see strong parallels in the mathematical framework used by these papers and ours. Both model nonlinear dynamics and approximate with a linearization around an operating point. Both study fold change dynamics.

One critical distinguishing feature is that we develop an approach for observability analysis of network models. If a network model were already available, we can use that model and apply observability analysis directly. Our data-driven observability maximization framework can be used to downselect biomarkers from a large set of genes for promoter selection, and is not used for network reconstruction. Because the scale of our network is large the network model is fundamentally unidentifiable from time-series RNA-seq data. RNA-seq costs prohibit us from collecting the volume of data that matches the scale of H. Xing et al., e.g., National Institutes of Health Gene Expression Omnibus ([http:// www.ncbi.nlm.nih.gov/geo/](http://www.ncbi.nlm.nih.gov/geo/)). Finally, our work completes the design-build-test-learn cycle in a way that was not economically possible in the early 2000s when the cited papers were published. In that way, this work critically validates that data-driven network modeling can discover novel, functional synthetic biological parts even in emerging organisms that lack extensive prior characterization.

Additionally, we strongly agree with the reviewer that we overlooked the field of network reconstruction as an important source of existing contributions to mode-of-action and sensor identification problems. To that end, we additionally cited Driscoll et al. (2006) Journal of Process Control and di Bernardo et al. (2005) Nature Biotechnology.

- The authors use the phrase “encoder genes”, but it is really the promoters of the genes that are responding. This is important as the genes themselves are not the biosensor elements. (Although a more traditional understanding of the definition of a biosensor would imply a protein that directly interacts with the analyte).

We thank the reviewer for calling to our attention the nomenclature issue. To address this comment from Reviewer 1 and a similar comment from Reviewer 2, we have replaced “encoder“ genes with ”biomarker“ genes or just “biomarkers” if appropriate and “observability-ranked” genes otherwise. We initially preferred “encoder” to denote genes ranked highly by our method as they can be viewed as genes which encode cell state information, however we agree that this can imply that the genes are biosensors which is not the case. However, if we expand the definition of a biosensor to be any biological component which reproducibly responds to an input with specificity, we may view the promoters of the genes as biosensors even if they are indirectly regulated in the presence of an input.

- If I understand correctly, the paper describes creating a “virtual sensor” by exposing each reporter promoter to the sample, and then mathematically aggregating the response. In other words, pooling is done at the data analysis level. How, in practice, could pooling be done at the assay level? This would be far more convenient.

The reviewer is correct. To address the lack of continuity in the manuscript due to this section (brought to our attention by Reviewer 2) we have removed the section on virtual sensing.

Pooling at the assay level could be done in several ways, however they each come with difficulties which led the authors to adopt the aggregation at the data analysis level.

One could develop a plasmid which contains k distinct sensor promoters driving expression of a unique reporter protein. Or one could drive k reporter proteins with the k distinct sensor promoters. Both of these approaches present the curse of dimensionality due to the combinatorial nature of combining k promoters from a set of N total promoters. Building and testing such strains can be very time-consuming and laborious.

One could also pool strains into a single culture to aggregate the responses of each reporter strain. However, the introduction of competition and interaction effects between strains is a confounding variable. In any case, we performed this experiment by pooling all 15 malathion reporters and inducing the consortium with malathion. We actually found this to be an effective strategy for malathion sensing and the results are shown in the revised manuscript in Figure 6. We thank the reviewer for calling such an experiment to our attention.

- To select encoder promoters, subsets of 15 random genes are selected from the top half of genes, as ranked by W . And then the best subset selected. Why this random approach. Why not simply select the top N genes ranked by W ?

We thank the reviewer for bringing this question to our attention. The randomized approach was explored and used for two reasons:

- i) the top genes, as shown in Figure 3d, are highly correlated, and
- ii) the sensor placement problem that maximizes the observability of the cell state is a relaxation of the actual sensor placement problem.

Though we are ranking genes optimally, since the sensor placement problem has been relaxed, the top N genes only approximately satisfy the reconstruction problem. Therefore, to avoid selecting correlated genes and maximize the cell state reconstruction accuracy, we have adopted the randomized selection approach motivated by entropy maximizing argument. The principle of maximum entropy states the probability distribution which best represents the current state of knowledge of the system is the one with largest entropy. For our system which has no prior experimental data on actual malathion sensing, the entropy maximizing distribution is the uniform distribution over the genes. We use this distribution for gene set selection by testing sets of genes for optimal reconstruction accuracy.

Other suitable approaches for gene selection would be to sort the genes from highest to lowest sampling weights, select the first gene to be in the library, then only select the next gene in the list for the library if it is dissimilar to all other genes in the library. One could explore the measure of dissimilarity to satisfy a metric such as orthogonality or Euclidean distance (if magnitude differences are of interest).

We have also now included a brief section in the Supplement that details the full sensor placement problem and what assumptions were made for the relaxed version.

- In the methods, it is initially confusing that W is in both the SVD decomposition and the matrix of estimated gene weights. It would be helpful to note somewhere near line 1115 that the W in this section is not the same as the W in the previous one.

Thanks for pointing out this issue. The right singular vectors from the SVD are now denoted by \mathbf{T} . The estimated gene weights remain as \mathbf{W} .

- There have been recently published manuscripts that describing successes in mining for novel biosensors that should be cited: J. Ibero et al. (2021) *Genes (Basel)*, C. Grazon et al. (2020) *Nature communications*

We have now cited both of the above publications. The first identifies genes partially responsible for the degradation of oestrogen through a growth protocol in oestrogen followed by genome sequencing. The second paper identifies candidates genes for progesterone sensing from RNA sequencing analysis. We thank the reviewer for pointing out the manuscripts.

Response to Reviewer 2

I have two main concerns:

1) *It is unclear what benefits this approach has over other methods to perform the same analysis. There is little mention or comparisons to other methods to cluster temporal RNA-seq data (e.g. hierarchical clustering based on a similarity score, graph-based clustering or community detection methods, use of bayes factors as gene-to-gene similarity scores; there are several others). Also, a similar analysis could be done with traditional differential expression analysis - I agree with authors that differential expression analysis was not conceived for temporal data, but most of the genes that their method selects (Fig 3D) are ultimately those are permanently up or down regulated after the stimulus, and I suspect these would have been picked up with differential expression analysis at T=8.*

This question is particularly relevant because the paper does use some of these same standard tools later in the paper: Fig 4b contains a hierarchical clustering of 15 genes and it is unclear why that cannot be extended to the original 624 genes. To me, it seems that a unique feature of their Gramian-based approach is the ability to rank genes according to importance (Figure 3A), but as I point out below, there are questions about the effectiveness of this approach.

To be clear, I don't think authors need to benchmark their approach to many other methods from the literature - I suggest instead they clearly highlight the benefits of their method. None of existing methods is perfect, so it would be valuable to clearly state the gaps covered by their method. Their approach is theoretically elegant and nicely combines tools from different disciplines but I suspect the broad readership of the journal will not necessarily appreciate the mathematical novelty and time series clustering is a very well covered subject.

We deeply appreciate the reviewer for bringing to our attention that we had not made clear the benefits of our developed approach. While one could identify dysregulated genes using differential expression analysis, there would be no guarantee that such an effect is temporally long-lived or if it is isolated to a short period of time. Especially since if differential expression analysis is the understood go-to approach, one would not collect a time-series RNA-seq dataset. With our approach, the entire profile is considered, and furthermore the “area under the curve” is considered in the observability maximization problem. This lets us identify genes from the time-series RNA-seq which may not have a large fold change with respect to the control at any one time point, but over the whole time-course the signal is significant. This is a capability lacking in the differential expression methods.

We agree with the reviewer that we missed out on emphasizing the benefits of our approach. Therefore, we have made clear in the introduction the advantage of our proposed approach over differential expression, clustering, and gene regulatory network reconstruction techniques (see L55-L68, L75-L89, and L104-L120). Furthermore, we have included a comparison of our approach and DESeq2 and it is now in Figure 3f and Supplementary Figure 9. We show that DESeq2 and our approach identify genes largely in complement to each other. We also highlight how DESeq2 is unable to call a large number of differentially expressed genes due to the low number of biological replicates.

2) I have an observation regarding the methodology itself. Fig 3A shows all 624 genes ranked by weights obtained from the Gramian approach. However, this curve looks almost linear, which unless I am missing something, casts doubts on the selectivity of the method. There is a comment on this in L435-437, but for the approach to be useful, one would expect an S-shaped curve so that a large number of genes have poor scores and few have a high score, and then genes can be selected by imposing a cut-off on the high scoring genes. Such almost linear relation suggests that the method is poorly selective and cannot discriminate between relevant and not relevant genes. I believe the panel on the right verifies this: for $T=8$ the three group of genes have very similar reconstruction accuracies; the text suggest that the high scorers (orange) are better but given the overlap between the violin plots, that claim should be backed up by a statistical significance test, e.g. by testing against a null hypotheses with randomly selected genes across all three groups. For $T=2$, there are interesting and clear differences across groups, and maybe this is the regime the authors should focus on, as it seems that the method does pick up differences in reconstruction power across the three groups of genes.

We thank the reviewer for their comments regarding the selectivity that our approach yields. Considering that time-series RNA-seq data is sparse and noisy, it is difficult to accurately reconstruct the network topology (gene-gene interactions). What we construct with DMD is a predictive model of the dynamics, the network topology is not fully captured. Moreover, DMD returns a fully-connected topology, i.e. all genes are influenced by nearly all other genes, albeit minorly. This leads to the situation in question, where the influence of important genes on observability are more difficult to separate from less influential genes. We have included a section in the Supplementary Information (Section 1.3), detailing the impact of the connectivity of the network on the learned sampling weights for a set of simulated systems. We show that for a fully connected network, the learned sampling weights are constant for each gene. As we decrease the connectivity, the weights more and more resemble an exponential curve. This explains the lack of an S-shaped curve in the previous draft of the manuscript.

To expose the differences between highly influential genes and non-influential genes, we make two modifications: i) taking the reviewers advice, we compare the groups of genes for short times ($T=2$) only – this makes sense when considering that for a fully-connected network, information can transfer exponentially fast from influential genes to non-influential genes, providing a non S-shaped curve, and ii) we normalize out any relationship between a gene’s sampling weight and a gene’s standard deviation – this results in a exponential shaped curve of the sampling weights and is shown in Fig. 3A of the revised manuscript. Dividing the gene sampling weights by their standard deviation, we now jointly can consider a confidence in the gene’s expression as well as the influence on observability.

As a note, as the cost of sequencing reduces and reconstruction of genetic networks of novel organisms becomes more routine, we anticipate that our methodology will prove extremely useful to optimal biomarker identification. From applications in drug discovery and biomanufacturing, targeted gene profiling can be used to greatly increase efficacy of therapeutics and efficiency of bioprocesses.

Specific comments:

- I am not sure if the term ‘encoder genes’ speaks to the audience. These would typically be called biomarkers, responsive genes, or similar.

We have incorporated the reviewers feedback and denote the encoder genes as biomarkers when appropriate and analyte-responsive genes or observability-ranked genes otherwise.

-
- L197-L200. Please explain how the 624 genes were selected; the normal approach is to use a volcano plot or similar to select differentially expressed genes (eg with DESeq2 or Limma) above a cutoff p -value; this is standard in rna-seq analysis.

The 624 genes were selected by first normalizing the raw transcript counts to obtain transcripts per million (TPM). Then two thresholds are set on the time-series for each gene: i) a mean cutoff of 100 TPM, and ii) a coefficient of variation threshold of 0.5 across replicates. The mean threshold is used

to filter out lowly expressed genes from which technical vs. biological noise can be difficult to assess (<https://genomebiology.biomedcentral.com/articles/10.1186/s13059-019-1861-6>). The coefficient of variation threshold allows us to quickly filter out genes with high noise levels.

We agree with the reviewer that using DESeq2 or Limma to assess whether a gene is differentially expressed is a standard approach for gene selection. However, we did not want to bias our observability maximization analysis by pre-filtering for only differentially expressed genes called by traditional analysis tools.

- *L204-207: Please explain how this assumption holds when the host metabolizes the analyte, i.e. its intracellular concentration cannot be assumed constant but decays in time.*

We thank the reviewer for pointing out this misguided statement. We agree that the malathion concentration must decay in time and therefore cannot be treated as a step input to the cell culture. This statement and references to impulse inputs have been removed from the manuscript.

- *L296-297 repeated in L279-280.*

Thanks for pointing out this mistake. We have removed the reference to the symmetry of oscillatory modes in L279-280.

- *L377-378. Please clarify what 'representative' stands for - does it mean there are only five clusters across the 624 genes?*

Thank you for pointing out this confusing statement. We have clarified in the text that we mean the predictive accuracy shown in the curves is representative of the predictive accuracy of most genes in the transcriptome.

- *L544-551: This is a key point but I could not understand how the 15 genes were selected and why there was a need for a randomization step. Why not select the top 15 genes ranked according to the computed sampling weights? This was done in L532-538 but it is unclear why a different procedure is needed here.*

We thank the reviewer for bringing this question to our attention. The randomized approach was explored and used for two reasons:

- i) the top genes, as shown in Figure 3d, are highly correlated, and
- ii) the sensor placement problem that maximizes the observability of the cell state is a relaxation of the actual sensor placement problem.

Though we are ranking genes optimally, since the sensor placement problem has been relaxed, the top N genes only approximately satisfy the reconstruction problem. Therefore, to avoid selecting correlated genes and maximize the cell state reconstruction accuracy, we have adopted the randomized selection approach motivated by entropy maximizing argument. The principle of maximum entropy states the probability distribution which best represents the current state of knowledge of the system is the one with largest entropy. For our system which has no prior experimental data on actual malathion sensing, the entropy maximizing distribution is the uniform distribution over the genes. We use this distribution for gene set selection by testing sets of genes for optimal reconstruction accuracy.

Other suitable approaches for gene selection would be to sort the genes from highest to lowest sampling weights, select the first gene to be in the library, then only select the next gene in the list for the library if it is dissimilar to all other genes in the library. One could explore the measure of dissimilarity to satisfy a metric such as orthogonality or Euclidean distance (if magnitude differences are of interest).

We have also now included a brief section in the Supplement that details the full sensor placement problem and what assumptions were made for the relaxed version.

- *L495-502: this is an insightful comment but unclear where this can be seen in Fig 3B.*

We thank the reviewer for their feedback. To clarify this point, we have added the following sentence after the aforementioned comment:

“Specifically, the reconstruction accuracy with 5 genes sampled for $T = 10$ time points is above 0.9 and the reconstruction accuracy with 600 genes sampled for $T = 8$ time points is just above 0.8.”

Note that the figure in question is no longer Fig 3B but is now Fig 3C.

- *L607-608: please clarify what is meant by sfGFP being a reporter for transcription initiation.*

We appreciate the reviewer pointing out the lack of clarity in the original manuscript. To clarify this point, the reporter protein sfGFP has a fast maturation time and long half-life. This implies that sfGFP accumulates inside the cell over long periods of time. If we consider a model for the protein concentration as $dp/dt = a*m - d*p$, we can consider the degradation rate to be miniscule and see that $dp/dt \approx a*m$, modeling this build-up of protein due to transcription initiation of the mRNA m . We hope this clarification is helpful.

- *L657: Michaelis-Menten kinetics are hyperbolic and do not resemble Hill equations (unless $n=1$).*

Thank you for pointing out this mistake. We have removed the reference to Michaelis-Menten kinetics and replaced it with Hill equation kinetics.

- *Figure 2A is unclear and think 2D should be the main content - I appreciate the effort to make the mathematics more accessible but I don't think end users will understand 2A. The Results does mention mathematical notation, but these are only defined in the Methods, which makes it hard to follow. I suggest to include few equations into the results section to make the text clear and unambiguous.*

Taking the reviewers advice, we have replaced Figure 2A with DMD prediction accuracy results, making the main focus of Figure 2 on the predictions. We have also included equations in the Results section to make the text more readable.

- *Suggest to merge Fig 2B-C because they share the data markers.*

Done – thanks for the suggestion!

- *In Figure 3A genes with highest rank have a score of 1, but a score of 0 in 3D.*

We have changed the ranking in 3D to reflect the ranking in 3A. Thank you for the clarifying point.

- *Figure 4B: were the gene-to-gene correlations computed at a fixed time, or averaged across all timepoints?*

The temporal response of the genes were correlated with each other. Additionally, this figure was removed from the manuscript as upon internal review, we consider it to be inconsequential to the message of our work.

• *Figure 4D: From the caption and Table 1 it seems that the dose-response curves were computed at different times, which were determined according to the maximal fluorescence. This is unusual and makes the dose-response curves not comparable to each other. Normally dose-response curves are computed at mid-exponential growth or similar. Moreover, without seeing the growth curves or kinetic data (sfGFP fluorescence/OD) it is impossible to establish if those selected times are representative or artefacts. It is also unclear if the maximal fold change mentioned in L667 is the ratio between sfGFP_sensor/sfGFP_control or (sfGFP_sensor/OD_sensor)/(sfGFP_control/OD_control). These issues can have a huge impact on the shape of the dose-response curve, see eg <https://pubs.acs.org/doi/abs/10.1021/acssynbio.2c00143>*

We thank the reviewer for this insightful comment and reference. We have displayed dose-response curves at different times as many reporters exhibit a transient response in their mid-log phase (Supplementary Figure 7). Displaying the dose-response curves at a fixed time point across all reporters will not accurately exhibit their distinct responses to malathion. Given that we sampled the transcriptome for 1.5 hours post-induction, we expect that some genes we've captured exhibit quick, transient responses. Reporter strains such as *acrA* and *lpxC* display an extremely acute response to malathion that quickly diminishes on the order of a few hours. Conversely reporter *rpoA* shows response after 12 hours of induction, due to stationary-phase activation. For these reasons, we have displayed the dose-response curves at different times. We have further provided the growth curves and kinetic data (sfGFP/OD) as supplementary figures (Supplementary Figures 5 and 6).

The maximal fold change is the ratio between (sfGFP_sensor/OD_sensor)/(sfGFP_control/OD_control).

• *Figure 4D: it would be nice to see the dynamic range and Km of all sensors in a single scatter plot; if done, it would be important to report error bars for uncertainty estimates given by the fitting procedure; particularly for the Hill coefficients as their fitted values are known to be very sensitive to noise.*

We have now included a scatter plot of Km vs. dynamic range and Km vs. Hill coefficient as Figure 4D in the manuscript. We have reported uncertainty estimates for Km and the Hill coefficient and have observed the noted sensitivity to noise for the Hill coefficients. Due to large amount of noise (orders of magnitude above the actual signal) for a few reporters, we shifted the transfer curve to be plotted at a time point only a few minutes from the time point used in the original manuscript. This resulted in a drastically reduced Hill coefficient for the reporter *sucC* from 21.6 to now 1.9.

• *Figures 5-6. I found these results slightly disconnected from the previous and at times read as if they could be the subject of a different paper. I suggest to improve the linkage between the two parts of the paper.*

Thanks for pointing out our lack of clarity with the latter sections of the paper. After reviewing the sections and their continuity with the main message of our work, we have decided to remove the section on virtual sensing from our revised manuscript. We have improved the linkage between the the early sections of the manuscript and the section regarding screening environmental samples for malathion. Note that we have also added two additional sections on specificity of the reporters and on pooling the reporters on an assay level. We hope that these modification, as well as the rewriting, has improved the clarity and continuity of the manuscript.

• *The computer code in github is well presented, but I'd encourage authors to either lock the repo, or provide the hash key of the commit that corresponds to the paper submission. Otherwise it is impossible to know which version of the code corresponds exactly to what was done in the manuscript. This is not their fault and is part of a wider trend in the field, where code repos are left open and, as a result, it is not possible to establish reproducibility. Another alternative is to simply attach code as a supplementary zip file.*

Thanks for pointing out this important issue. We have provided the hash key of the github repo upon resubmission (2aaa256)

Reviewers' Comments:

Reviewer #1:

Remarks to the Author:

In the revised manuscript and response document, the authors have done a nice job of responding to all the concerns raised in my initial review. Overall the manuscript is significantly improved. However, I do have a few remaining concerns that I would prefer to see addressed

1. My primary concern with the initial manuscript was the assertion that the author's method identifies biosensors of a compound rather than biomarkers of the cellular response to a compound. The authors have acknowledged this important distinction and have done a good job revising the text of the manuscript in response. I would note that on line 20 of the abstract, the phrase "library of living melathione sensors" is not quite accurate. There is a single living melathione sensor which is the host cell – the individual promoters are readouts on this whole cell biosensor. Apart from this line, I am satisfied with the authors response to this concern.

2. I had also previously raised a concern about the method of randomly selecting from the gene (promoters) ranked by their observability weights. The authors also responded in depth to this question, but their answer does not fully satisfy my concerns. I understand the authors rationale that there are significant correlations among the ranked genes. I think that describing this in the text strengthens the manuscript.

However, the randomization approach continues to raise questions about the value of the ranking by weights. Specifically, the authors rank all ~600 genes by rank, and then randomly select 15 genes from the *top half (~300) of the ranked set*. This is equivalent to choosing the best 15 genes from 50% of all possible genes regardless of the rank within this top 50%. Or, in other words, the role of the weights was to simply triage 50% of the promoters.

The question is whether the ranking added significant value to this random selection apart from the 50% triage (especially given that *acrA* was precisely at the 50% cutoff – Figure 3e). One easy way to address this is to choose the top 15% genes from *all* genes. Would this have resulted in at least as good a set? What if one selected 15 random promoters using the Monte Carlo sampling from a random set of 300 promoters (thus ensuring the sampling works on a candidate set of the same size).

Regarding correlations among the selected promoters, the authors do add a new analysis showing that the overall correlation of the selected 15 is lower than the top weighted 15. This is nice. But there is certainly still correlation between the 15 selected. (C.f. Figure 4c of the revised manuscript, many of the 15 selected genes show highly similar responses).

If selecting uncorrelated promoters is the crucial point of the random sampling, it would be valuable to perform the selection using the alternative approach described by the authors in the revision: "select genes from the top of the ranking and remove any genes 537 which are correlated until only 15 genes remain". I am not suggesting redoing all the experiments with a set determined this way – but a supplemental figure showing the genes that would have resulted, and how they compare to the actual 15 might help shed light on this lingering question.

3. The authors have added new experiments and text addressing my concern about the specificity of the identified biomarkers. The new data do strengthen the paper. But I believe some limitations of these data need to be acknowledged.

It is worth noting that comparing responses to 4 other compounds is not sufficient to claim that the response is unique to malathione. At best, the biosensors are not responsive to these 4 other genes.

More importantly, it is not clear that the other 4 compounds selected represent the most likely candidates for cross-reactivity. For example, one would expect that compounds that have the same cellular effect as malathione will induce similar responses. This might include compounds

with similar structures (zeta-cypermethrin and permethrin are structurally very similar to each other but less so to malathione). Also, some Pseudomonads are thought to degrade malathione (Goda (2010) Biodegradation). If this is the case, metabolically related compounds (e.g. naphthol) would be prime candidates.

I certainly understand the practical limitations of experimentally testing all possible confounding compounds. I suggest at least modify the text to acknowledge the limitations of the experiment on just the 4 compounds selected.

4. The authors acknowledged the concern with virtual biosensor approach in the original manuscript and have replaced this with an experiment describing experimental pooling. This is a welcome improvement to the manuscript.

I have one question regarding these new data: is the temporal response of the pooled reporters (Figure 6) what would have been predicted from the response of the individual reporters (Figure 4b)?

By eye it is not clear that this is the case. The pooled reporter response continues increasing beyond 20 hours. From Figure 4b, it does not seem that any of the individual reporters show this pattern. I would have expected (perhaps incorrectly) an increase then plateau or even drop based on visually summing Figure 4b.

The underlying concern is whether the temporal response reflects solely the response of the reporter promoters (and thus might be expected to sum), or perhaps temporal dynamics of growth and competition between the individual strains (which could change the relative contributions of each strain over time).

5. The authors include new data comparing their approach to promoter selection, with an approach based on using the well established DESeq2 program. This is welcome data. However, I am concerned by the overall lack of agreement between the two methods. I am particularly concerned by their statement that their "approach identifies genes almost exactly where DESeq2 identifies no genes to be differentially expressed". At face value this is troubling because it means that the authors approach uses genes that are not statistically significantly differentially expressed (by DESeq2). The authors suggest that this illustrates an advantage of their approach, but it is not obvious why. This result deserves more discussion than it currently receives in the revised manuscript.

6. The other concerns I raised have been adequately addressed.

Reviewer #2:

Remarks to the Author:

I thank the authors for their comprehensive revision and was glad to see that most feedback has been taken on board. I have two remarks:

1) Randomized selection of top responders: both reviewers commented on this key issue. In my opinion, there is no need to do that random step, and to some extent it throws away the core principle of their method (which is quite elegant).

The rebuttal argues this was needed because of top responders being highly correlated. While this seems to be true judging from Fig 3D (though these are z-score normalized so correlations cannot be directly inferred from them), I could not find a figure showing the `_ranked_` top responders are also correlated. E.g. is it possible the top responder has an upward response and the second top response a slightly different response? If that's the case then there is no reason to do the random step. I suggest authors examine those gene-to-gene correlations carefully, eg by looking at a gene-to-gene correlation heatmap per time point, or using correlation scores between whole time series.

That being said, I do not think those correlations are necessarily a problem or take away from the contribution. E.g. if the method identifies 20 top genes that respond a similar way but are mechanistically disconnected (eg from different pathways), then they can be useful by themselves. On the contrary, if the method was finding 20 highly correlated genes that are known to interact with one another, then that would be a potential issue.

There is also a comment on an entropy maximization argument, but this is strange because in principle one could apply the same argument and the uniform sampling to the whole geneset without using their method to preselect a subset of genes.

2) Dose-response curves in Fig 4C. I agree with the argument in the rebuttal but feel this needs to be clearer in the caption, y-axis label, and Suppl Table 4 to clearly say that this is maximal fluorescence. This is key because one can easily choose another criterion to select which timepoints to plot and easily distort the shape of the resulting dose-response curve.

Also, there was a comment in the rebuttal

“Due to large amount of noise (orders of magnitude above the actual signal) for a few reporters, we shifted the transfer curve to be plotted at a time point only a few minutes from the time point used in the original manuscript. This resulted in a drastically reduced Hill coefficient for the reporter sucC from 21.6 to now 1.9.”

Does this mean that the dose-response curve for sucC was reported at a slightly different timepoint, and not the one of maximal fluorescence? Please clarify.

Response to reviewer comments

Aqib Hasnain et al.

Response to Reviewer 1

In the revised manuscript and response document, the authors have done a nice job of responding to all the concerns raised in my initial review. Overall the manuscript is significantly improved. However, I do have a few remaining concerns that I would prefer to see addressed

We would like to earnestly thank the reviewer for their valuable feedback on the original manuscript and for providing additional thorough feedback on the revisions.

1. My primary concern with the initial manuscript was the assertion that the author's method identifies biosensors of a compound rather than biomarkers of the cellular response to a compound. The authors have acknowledged this important distinction and have done a good job revising the text of the manuscript in response. I would note that on line 20 of the abstract, the phrase "library of living melathione sensors" is not quite accurate. There is a single living melathione sensor which is the host cell – the individual promoters are readouts on this whole cell biosensor. Apart from this line, I am satisfied with the authors response to this concern.

Thank you for pointing out this lingering inaccuracy. We have revised this statement in the abstract to "The engineered host cell, a living malathion sensor" (L19-L20).

2. I had also previously raised a concern about the method of randomly selecting from the gene (promoters) ranked by their observability weights. The authors also responded in depth to this question, but their answer does not fully satisfy my concerns. I understand the authors rationale that there are significant correlations among the ranked genes. I think that describing this in the text strengthens the manuscript.

*However, the randomization approach continues to raise questions about the value of the ranking by weights. Specifically, the authors rank all 600 genes by rank, and then randomly select 15 genes from the *top half (300) of the ranked set*. This is equivalent to choosing the best 15 genes from 50% of all possible genes regardless of the rank within this top 50%. Or, in other words, the role of the weights was to simply triage 50% of the promoters.*

*The question is whether the ranking added significant value to this random selection apart fro the 50% triage (especially given that *acrA* was precisely at the 50% cutoff – Figure 3e). One easy way to address this is to choose the top 15% genes from *all* genes. Would this have resulted in at least as good a set? What if one selected 15 random promoters using the Monte Carlo sampling from a random set of 300 promoters (thus ensuring the sampling works on a candidate set of the same size).*

We thank the reviewer for their suggestions on how to address the value of observability-based ranking. We have implemented both of the suggestions: a) selecting the top 15% of genes from all genes and sampling from this set, b) selecting random sets of 300 genes and sampling from this set. What we find for a) is that the chosen gene set ranks in the top 95 percentile for reconstruction accuracy and is the top set for the correlation metric. What we find for b) is that the chosen gene set is slightly outperformed in both reconstruction accuracy and correlation metric by only a single other gene set. The results are visualized in Supplementary Figure 7 (a) and (b), respectively. One conclusion that can be made then is that the chosen gene set provides balance between the degree of uncorrelated response and observability.

We highlight in the manuscript that multiple strategies can be employed to select genes given the ranking (Lines 530-534). In our case, the diversity of reporter response was important enough that we chose to use the Monte Carlo approach for gene selection. However, if 15 top responders are desired, then one can choose the top 15 genes by rank. We show that the top 15 genes are highly correlated in Supplementary Figure 5b, and in Supplementary Figure 5c we show that these top responders belong to a diverse set of biological processes, which can be useful for interrogating broad sets of processes with a small library.

Regarding correlations among the selected promoters, the authors do add a new analysis showing that the overall correlation of the selected 15 is lower than the top weighted 15. This is nice. But there is certainly still correlation between the 15 selected. (C.f. Figure 4c of the revised manuscript, many of the 15 selected genes show highly similar responses).

If selecting uncorrelated promoters is the crucial point of the random sampling, it would be valuable to perform the selection using the alternative approach described by the authors in the revision: "select genes from the top of the ranking and remove any genes 537 which are correlated until only 15 genes remain". I am not suggesting redoing all the experiments with a set determined this way – but a supplemental figure showing the genes that would have resulted, and how they compare to the actual 15 might help shed light on this lingering question.

The authors thank the reviewer for providing critical feedback on their methodology. In the revised manuscript we detail the considerations for gene selection using the learned observability ranking; namely, 1) gene rank, 2) correlations, and 3) reconstruction accuracy (contribution to observability). As suggested by the reviewer, we also employ the correlation-based strategy for selecting genes and display them in Supplementary Figure 6. We compute the correlation metric (eq (4) in revised manuscript) and reconstruction accuracy for gene sets chosen with each criteria and show that for 1) reconstruction accuracy is moderate and correlations are high, for 2) reconstruction accuracy is low and correlations are low, and for 3) reconstruction accuracy is high and correlations are moderate. This provides the rationale for selecting genes via the third criteria (i.e, the randomized approach). The revisions are made in Lines 530-579.

3. The authors have added new experiments and text addressing my concern about the specificity of the identified biomarkers. The new data do strengthen the paper. But I believe some limitations of these data need to be acknowledged.

It is worth noting that comparing responses to 4 other compounds is not sufficient to claim that the response is unique to malathione. At best, the biosensors are not responsive to these 4 other compounds.

More importantly, it is not clear that the other 4 compounds selected represent the most likely candidates for cross-reactivity. For example, one would expect that compounds that have the same cellular effect as malathione will induce similar responses. This might include compounds with similar structures (zeta-cypermethrin and permethrin are structurally very similar to each other but less so to malathione). Also, some Pseudomonads are thought to degrade malathione (Goda (2010) Biodegradation). If this is the case, metabolically related compounds (e.g. naphthol) would be prime candidates.

I certainly understand the practical limitations of experimentally testing all possible confounding compounds. I suggest at least modify the text to acknowledge the limitations of the experiment on just the 4 compounds selected.

We appreciate the reviewers feedback on the specificity experiments of the identified biomarkers. We agree that comparing the response to malathion with the response to 4 other compounds does not certify that the biomarkers are specifically sensitive to malathion.

We have now made the limitations of our experiments clear in the manuscript, see Lines 788 - 793. We thank the reviewer for also pointing out the molecule naphthol as a metabolically related compound to malathion. This can prove useful for any future work comprising environmental testing.

4. The authors acknowledged the concern with virtual biosensor approach in the original manuscript and have replaced this with an experiment describing experimental pooling. This is a welcome improvement to the manuscript.

I have one question regarding these new data: is the temporal response of the pooled reporters (Figure 6) what would have been predicted from the response of the individual reporters (Figure 4b)?

By eye it is not clear that this is the case. The pooled reporter response continues increasing beyond 20 hours. From Figure 4b, it does not seem that any of the individual reporters show this pattern. I would have expected (perhaps incorrectly) an increase than plateau or even drop based on visually summing Figure 4b.

The underlying concern is whether the temporal response reflects solely the response of the reporter promoters (and thus might be expected to sum), or perhaps temporal dynamics of growth and competition between the individual strains (which could change the relative contributions of each strain over time).

We appreciate and welcome the question about the experimental pooling of the reporters. In Figure 4b, we do see only a few of the reporters that show a response up to 20 hours of growth. Specifically, the reporters are anti-sigma 28 factor, Uncharacterized protein I, and cspA2. The sustained signal may be more clearly demonstrated in Supplementary Figure 7 which shows the response of the OD normalized sfGFP fluorescence, rather than the fold change of this with respect to the negative control.

The reviewers underlying concern is noted. The individual reporters, when pooled together, almost certainly face growth competition and the relative proportions of each strain in the pool is subject to change over time. Despite this competition effect, we do see a salient response to malathion with a higher sustained fold change than any individual reporter. One could conceive of optimizing the initial proportions of each strain to design a pooled reporter with desired characteristics; higher fold change, lower biological/technical noise, distinct responses to distinct input profiles (e.g. step input of malathion versus linear ramp input of malathion). This would be a useful biotechnological study describing a novel approach to detection and we consider it as future direction for the reporter library.

5. The authors include new data comparing their approach to promoter selection, with an approach based on using the well established DESeq2 program. This is welcome data. However, I am concerned by the overall lack of agreement between the two methods. I am particularly concerned by their statement that their “approach identifies genes almost exactly where DESeq2 identifies no genes to be differentially expressed”. At face value this is troubling because it means that the authors approach uses genes that are not statistically significantly differentially expressed (by DESeq2). The authors suggest that this illustrates an advantage of their approach, but it is not obvious why. This result deserves more discussion that it currently receives in the revised manuscript.

We thank the reviewer for their feedback on the differential expression analysis. It is important that we mention that DESeq2 identifies a very small number of genes to be statistically significantly expressed. Notably, only 5 are called significantly differentially expressed after multiple testing corrections of the p-values. This is significant as we then claim that an advantage of our approach is to be able to identify genes which DESeq2 does not. Moreover, these genes are clustered (by correlation) into two groups, one group of size 3 and one group of size 2. Of these 5 genes, two are ranked in the top 82% by observability maximization and 4 in the top 55%, so there is considerable overlap in the statistically significant differentially expressed genes and our approach. Given these results, we suspect that as the number of biological replicates increases, our approach and differential expression analysis will tend to converge.

The 180 genes that we use to compare to our approach are the 5 genes mentioned above plus the 175 genes out of 6009 that have non-corrected p-values less than 0.05. Of these 180, it turned out that the genes identified by our approach and DESeq2 were largely in complement to each other. We find this to be important, and since we validated our approach by building a reporter library with genes not identified by DESeq2, we consider this analysis to show our method to be useful when analyzing time-course RNA-seq data.

We have now removed the statement “our approach identifies genes almost exactly where DESeq2 identifies no genes to be differentially expressed” as to avoid placing too strong of a conclusion from the analysis. Furthermore, we have emphasized that the majority of genes from DESeq2 are themselves not statistically significantly differentially expressed (see lines 620-630). We have also made a note of this in Fig 3F.

6. The other concerns I raised have been adequately addressed.

The authors have noted and appreciated this statement and again thank the reviewer for providing valuable feedback to improve our manuscript.

Response to Reviewer 2

I thank the authors for their comprehensive revision and was glad to see that most feedback has been taken on board. I have two remarks:

We sincerely thank the reviewer for their valuable feedback on the original manuscript and for providing additional feedback on the revisions.

1) Randomized selection of top responders: both reviewers commented on this key issue. In my opinion, there is no need to do that random step, and to some extent it throws away the core principle of their method (which is quite elegant).

The rebuttal argues this was needed because of top responders being highly correlated. While this seems to be true judging from Fig 3D (though these are z-score normalized so correlations cannot be directly inferred from them), I could not find a figure showing the ranked top responders are also correlated. E.g. is it possible the top responder has an upward response and the second top response a slightly different response? If that's the case then there is no reason to do the random step. I suggest authors examine those gene-to-gene correlations carefully, eg by looking at a gene-to-gene correlation heatmap per time point, or using correlation scores between whole time series.

That being said, I do not think those correlations are necessarily a problem or take away from the contribution. E.g. if the method identifies 20 top genes that respond a similar way but are mechanistically disconnected (eg from different pathways), then they can be useful by themselves. On the contrary, if the method was finding 20 highly correlated genes that are known to interact with one another, then that would be a potential issue.

We thank the reviewer for the feedback and suggestions. To show that the ranked top responders are heavily correlated, we have now included the correlation heatmap in Supplementary Figure 5b. We can see that among the top 15 responders, two positively associated clusters are apparent. If we neglect signs, then there is only a single cluster in which the 15 responders can fit into. On the contrary, for the 15 genes that were chosen for the reporter library, three clusters emerge. These correlations were computed between whole time-series.

The final comment in the point above by the reviewer points out that the correlations may not be an issue, particularly if the responders are mechanistically disconnected. We show in Supplementary Figure 5 (panel (c)) that, although a large fraction of genes code for constituents of a ribosomal subunit, overall they belong to mechanistically distinct pathways.

In the manuscript (Lines 530-579) we have detailed the considerations for gene selection using the observability-based ranking and emphasized that the Monte Carlo based selection strategy is one approach for gene selection. We have provided another Supplementary Figure 6 demonstrating the genes that would be selected given the correlation-based strategy for selection. This strategy starts by selecting the gene that was ranked highest, then keeping the next gene in the library only if it passes a correlation cutoff.

We state that if responsiveness is the main criteria for gene selection, then choosing the top responders may be more appropriate. However, if diversity of reporter responses is of interest, than a correlation-based strategy or Monte Carlo based strategy may be more suitable.

There is also a comment on an entropy maximization argument, but this is strange because in principle one could apply the same argument and the uniform sampling to the whole geneset without using their method to preselect a subset of genes.

We thank the reviewer for their comment and we agree that the entropy maximization argument can be applied to the entire set of genes. However, this would take into account genes that would not contribute to observability at all or very little. Then this would have the effect of sampling genes that we predict would not respond to malathion after constructing genetic reporters from their promoters.

2) Dose-response curves in Fig 4C. I agree with the argument in the rebuttal but feel this needs to be clearer in the caption, y-axis label, and Supplementary Table 4 to clearly say that this is maximal fluorescence. This is key because one can easily choose another criterion to select which timepoints to plot and easily distort the shape of the resulting dose-response curve.

We thank the reviewer for the valuable suggestion and have implemented it in the caption and y-axis label of Fig 4C and Supplementary Table 4.

*Also, there was a comment in the rebuttal “Due to large amount of noise (orders of magnitude above the actual signal) for a few reporters, we shifted the transfer curve to be plotted at a time point only a few minutes from the time point used in the original manuscript. This resulted in a drastically reduced Hill coefficient for the reporter *sucC* from 21.6 to now 1.9.”*

*Does this mean that the dose-response curve for *sucC* was reported at a slightly different timepoint, and not the one of maximal fluorescence? Please clarify.*

We apologize for any confusion that we may have introduced with our statement about noise for a few of the reporters. To be clear, the noise and signal being referred to are the uncertainty estimate of best-fit parameters and the best-fit parameters themselves of the Hill function, respectively. We found that for the time point of maximal fold change for reporter *sucC*, the uncertainties were large. Therefore, we decided to shift the time point by one sample (3 minute sampling interval) which resulted in low uncertainty in the best-fit parameters. The maximal fold change before the shift was 1.30 ± 0.06 and after the shift is 1.27 ± 0.04 (mean and standard deviation across three replicates), i.e., within margin of noise of one another.

Reviewers' Comments:

Reviewer #1:

Remarks to the Author:

The authors have done a commendable job of responding to concerns raised during the entirety of the review process. The latest changes and responses do a good job of addressing the questions raised in the previous version.

The new supplemental figures, especially Figure 7, are very helpful for the reader to understand the strengths and limitations of the approach described.

With respect to the concern raised about cell growth dynamics during pooling, I agree with the authors response: 'The individual reporters, when pooled together, almost certainly face growth competition and the relative proportions of each strain in the pool is subject to change over time. Despite this competition effect, we do see a salient response to malathion with a higher sustained fold change than any individual reporter'. It is not clear to me if this has been noted in the revised manuscript, but I suggest that this point be included.

Apart from this one small point, I have no other concerns. I believe the manuscript is well-suited for publication in Nature Comm.

Reviewer #2:

Remarks to the Author:

My comments have been addressed satisfactorily. I thank the authors for their patience and willingness to engage constructively in the peer review process.

Response to reviewer comments

Aqib Hasnain et al.

Response to Reviewer 1

The authors have done a commendable job of responding to concerns raised during the entirety of the review process. The latest changes and responses do a good job of addressing the questions raised in the previous version.

The new supplemental figures, especially Figure 7, are very helpful for the reader to understand the strengths and limitations of the approach described.

We thank the reviewer for their constructive criticism to better our manuscript throughout the review process.

With respect to the concern raised about cell growth dynamics during pooling, I agree with the authors response: 'The individual reporters, when pooled together, almost certainly face growth competition and the relative proportions of each strain in the pool is subject to change over time. Despite this competition effect, we do see a salient response to malathion with a higher sustained fold change than any individual reporter'. It is not clear to me if this has been noted in the revised manuscript, but I suggest that this point be included.

We have now included this commentary in the manuscript. Previously we had only described the saliency of response, we now describe growth competition effects.

Apart from this one small point, I have no other concerns. I believe the manuscript is well-suited for publication in Nature Comm.

Thanks again, we appreciate your comment.

Response to Reviewer 2

My comments have been addressed satisfactorily. I thank the authors for their patience and willingness to engage constructively in the peer review process.

We thank the reviewer for their valuable feedback on the manuscript throughout the review process.